# Structures of wild-type and a constitutively closed mutant of connexin26 shed light on channel regulation by $CO_2$

Deborah H Brotherton[1], Sarbjit Nijjar[1], Christos G Savva[2], Nicholas Dale[1]*, Alexander David Cameron[1]*

[1]School of Life Sciences, University of Warwick, Coventry, United Kingdom; [2]Leicester Institute of Structural and Chemical Biology, Department of Molecular and Cell Biology, University of Leicester, Leicester, United Kingdom

**\*For correspondence:**
n.e.dale@warwick.ac.uk (ND);
a.cameron@warwick.ac.uk (ADC)

**Competing interest:** The authors declare that no competing interests exist.

**Abstract** Connexins allow intercellular communication by forming gap junction channels (GJCs) between juxtaposed cells. Connexin26 (Cx26) can be regulated directly by $CO_2$. This is proposed to be mediated through carbamylation of K125. We show that mutating K125 to glutamate, mimicking the negative charge of carbamylation, causes Cx26 GJCs to be constitutively closed. Through cryo-EM we observe that the K125E mutation pushes a conformational equilibrium towards the channel having a constricted pore entrance, similar to effects seen on raising the partial pressure of $CO_2$. In previous structures of connexins, the cytoplasmic loop, important in regulation and where K125 is located, is disordered. Through further cryo-EM studies we trap distinct states of Cx26 and observe density for the cytoplasmic loop. The interplay between the position of this loop, the conformations of the transmembrane helices and the position of the N-terminal helix, which controls the aperture to the pore, provides a mechanism for regulation.

## eLife assessment

This study presents **valuable** new structures of a carbamylation-mimetic K125E mutant of the Cx26 gap junction channel uncovering the cytoplasmic loop structure and information about the closed state of the channel. The cryo-EM maps are in high quality and serve as strong foundations for dissecting the gating mechanism by CO2, providing **convincing** evidence in support of a mechanism where CO2-mediated carbamylation of Lys125 shifts the conformational equilibrium towards a state where the N-terminus occludes the pore of the channel. This information will be of interest to biochemists, cell biologists and biophysicists interested in the function of gap-junction channels in health and disease.

## Introduction

Connexins form hexameric channels in the plasma membrane known as hemichannels, which can either function as regulated passageways between the cell and its environment, or dock with a hemichannel from another cell to form a dodecameric intercellular channel, or gap junction channel (GJC). Connexins have been shown to be directly regulated by various stimuli such as voltage (*Valiunas, 2002*; *Young and Peracchia, 2004*), pH (*Bevans and Harris, 1999*; *Khan et al., 2020*; *Yu et al., 2007*) or indirectly via intracellular calcium ion concentrations (*Peracchia, 2004*). Recent reports based on structural data also suggest that lipids may be involved in regulation (*Lee et al., 2023a*; *Lee et al.,*

2023b; Qi et al., 2023a). We have shown, however, that connexin26 (Cx26) and other similar β-connexins (Cx30, Cx32) can be regulated by the direct action of physiological concentrations of carbon dioxide independently of pH (Huckstepp et al., 2010; Meigh et al., 2013). Mutants of Cx26 are a leading cause of congenital deafness (Xu and Nicholson, 2013). While many of the mutations are non-syndromic, others lead to severe diseases such as keratitis ichthyosis deafness syndrome (KIDS) (Xu and Nicholson, 2013). Hemichannels are known to have different properties to GJCs (Stout et al., 2004) and our previous results show that an increase in the partial pressure of $CO_2$ ($PCO_2$) will open Cx26 hemichannels (Huckstepp et al., 2010) but close Cx26 GJCs (Nijjar et al., 2021), which are open under physiological conditions of $PCO_2$.

There are 20 connexin genes in the human genome (Abascal and Zardoya, 2013) and several structures have now been published (Bennett et al., 2016; Brotherton et al., 2022; Flores et al., 2020; Khan et al., 2020; Lee et al., 2023a; Lee et al., 2020; Lee et al., 2023b; Maeda et al., 2009; Myers et al., 2018; Qi et al., 2023a; Qi et al., 2023b). The connexin subunit, which is common to all, consists of four transmembrane helices (TMs) with a cytoplasmic N-terminal helix that in the hexameric arrangement of the hemichannel points towards the central pore (Maeda et al., 2009). In structures of the dodecameric GJC, the extracellular part involved in docking is well defined, whereas the cytoplasmic region is much more variable. A large cytoplasmic loop between TM2 and TM3, shown to be involved in regulation, has not been visible in any structure. Structures of the connexins either have the N-terminal helices tucked back against the wall of the channel (Flores et al., 2020; Lee et al., 2023a; Lee et al., 2023b; Myers et al., 2018), in a raised position (Lee et al., 2023a; Lee et al., 2020; Qi et al., 2023a), in an intermediate position (Brotherton et al., 2022; Lee et al., 2023a; Maeda et al., 2009), or not well defined in the density (Bennett et al., 2016; Brotherton et al., 2022; Khan et al., 2020; Lee et al., 2023b; Qi et al., 2023b). The position of the N-terminal helix is thought to be important in the regulation of channel permeability. We have shown previously for human Cx26 GJCs, that the position of the N-terminus is dependent on $PCO_2$ at constant pH (Brotherton et al., 2022). By examining structures from protein vitrified at different levels of $PCO_2$, we observed that under conditions of high $PCO_2$ the conformation of the protein was biased towards a conformation where the N-terminus protrudes radially into the pore to form a constriction at the centre ($N_{Const}$). Two distinct conformations of the protein were seen with the predominant difference between them in the cytoplasmic portion of TM2 (Brotherton et al., 2022) where an anticlockwise rotation of TM2 correlated with more definition of the density for the N-terminus. On the other hand, under low $PCO_2$ conditions the conformation with the more defined N-terminus was not observed and the channel appeared more open ($N_{Flex}$).

Based on a wealth of mutational data it has been hypothesised that the regulation of hemichannel opening by $CO_2$ is through a carbamylation reaction of a specific lysine (Meigh et al., 2015; Meigh et al., 2013; Nijjar et al., 2021). This post-translational modification is a reversible and highly labile reaction of $CO_2$ (Lorimer, 1983) that effectively changes the charge of a neutral lysine residue to make it negative. A so-called 'carbamylation motif' was identified in $CO_2$-sensitive connexins (Dospinescu et al., 2019; Meigh et al., 2013) that when introduced into a related $CO_2$-insensitive connexin rendered the protein $CO_2$-sensitive (Meigh et al., 2013). In Cx26 this motif has the sequence $K_{125}VRIEG_{130}$. In the crystal structure of the Cx26 GJC that was published in 2009 (Maeda et al., 2009), Lys125, which is conserved amongst β-connexins that are known to be modulated by $CO_2$ (Dospinescu et al., 2019), is positioned near to the N-terminus of TM3 within ~6 Å of Arg104 of TM2 of the neighbouring subunit, at either side of the disordered cytoplasmic loop. It was suggested that upon carbamylation, the negative charge of the modified lysine would attract Arg104 causing a conformational change (Meigh et al., 2013). In hemichannels, mutation of Lys125 to glutamate, so mimicking the charge of the carbamylated lysine (K125E), results in constitutively open hemichannels consistent with elevated $PCO_2$, whereas the corresponding K125R mutation results in hemichannels that cannot be opened by $CO_2$ (Meigh et al., 2013). In GJCs, the K125R mutation results in the protein not closing in response to $CO_2$, though importantly, this mutation does not prevent closure by acidification (Nijjar et al., 2021).

While our previous structures (Brotherton et al., 2022) demonstrated an effect of $PCO_2$ on the conformation of the protein, neither Lys125 nor Arg104 were visible in the density. Here, we probe this further. By solubilising the protein in the detergent lauryl maltose neopentyl glycol (LMNG) rather than dodecyl β-D-maltoside (DDM), we obtain much improved density for the cytoplasmic region of

**Table 1.** Cryo-EM data collection and processing statistics.

| | K125E$_{90}$ | K125R$_{90}$ | LMNG$_{90}$ | K125E$_{HEPES}$ | WT$_{HEPES}$ |
|---|---|---|---|---|---|
| Voltage (kV) | 300 | 300 | 300 | 300 | 300 |
| Magnification (×1000) | 105 | 105 | 105 | 75 | 75 |
| Camera | K3 | K3 | K3 | Falcon 3 | Falcon 3 |
| Frame alignment on Falcon 3 | – | | – | Yes | Yes |
| Camera mode | Super-resolution | Counting bin 1 | Super-resolution bin 2 | Counting | Counting |
| Energy filter (eV) | 20 | 20 | 20 | 20 | 20 |
| Defocus range (μm) | –0.8 to –2.0 | –0.8 to –2.0 | –0.8 to –2.3 | –0.3 to –1.7 | –0.5 with Volta phase-plate |
| Pixel size (Å/pix) | 0.85 | 0.835 | 0.835 | 1.08 | 1.08 |
| Dose on detector (e$^-$/pix/s) | 10 | 15 | 18.77 | | 0.7 |
| Dose on sample (e$^-$/pix/s) | 11.4 | 18 | | 1.06 | 0.69 |
| Exposure time (s) | 3 | 2 | 2 | 44.01 | 60 |
| No. of images | 4731 | 10,044 | 11,362 | 2573 | 2436 |
| Frames per image | 45 | 50 | 50 | 40 | 75 |
| Final particle number | 189,887 | 222,622 | 204,438 | 147,546 | 60,995 |
| Resolution* | | | | | |
| Masked D6 (Å) | 2.2 | 2.1 | 2.0 | 4.3 | 4.9 |

*From Relion_postprocess (**Scheres, 2012**).

the protein. We refine two conformationally diverse structures of the protein, where we see much more defined differences in the cytoplasmic region of the protein than we were able to observe previously. These data suggest a mechanism for closure involving the concerted movements of TM2 and the KVRIEG motif. We show that the structure with the K125E mutation matches the more closed conformation of the protein.

## Results

### Conformations of TMs 1, 2 and the KVRIEG motif correlate with position of the N-terminus

In an attempt to improve the resolution of the cytoplasmic region of the protein, we changed the method of solubilisation and purification by substituting DDM with LMNG and swapping from phosphate buffers to $CO_2$/$HCO_3^-$ buffers throughout the process. The switch from DDM to LMNG was intended to provide clarity on the provenance of density within the pore of the protein that was observed in our previous structures (*Brotherton et al., 2022*) and which was considered to be either DDM, as an artefact of solubilisation, or lipid. Lipids have been observed in the pore of other connexins and related large-pored channels and have been suggested to be part of the mechanism (*Burendei et al., 2020*; *Lee et al., 2023a*; *Lee et al., 2020*; *Lee et al., 2023b*). Our use of high $CO_2$/$HCO_3^-$ buffers throughout purification was intended to keep the gap junction in the closed state throughout, and hence reduce the chance of extraneous lipid or detergent entering the channel pore.

Data were collected from Cx26 vitrified at a $PCO_2$ corresponding to 90 mmHg. Refinement with D6 symmetry imposed resulted in a map with a nominal resolution of 2.0 Å as defined by gold standard Fourier shell correlations (FSC) (*Rosenthal and Henderson, 2003*; *Scheres, 2012*; *Figure 1—figure supplement 1*, *Table 1*). This was further classified using the procedure that we previously developed, involving particle expansion and signal subtraction, to focus on just the cytoplasmic region of one of the two docked hemichannels (*Figure 1—figure supplements 1 and 2*; *Brotherton et al., 2022*).

The results from this classification were broadly in line with our previous results. However, improved definition of the density in the cytoplasmic region enabled us to model this region more accurately. As before, the position of the cytoplasmic region of TM2 in these maps correlated with the presence or absence of the N-terminus. Maps from two classifications based on the most extreme positions of TM2 and corresponding clarity of the N-terminus were taken forward for further analysis (*Figure 1—figure supplement 1*, *Table 2*). These maps, both of which have a similar resolution of 2.3 Å, have been respectively denoted by LMNG-N$_{Const}$ (N-terminus defined and constricting the pore) and LMNG-N$_{Flex}$ (N-terminus not visible) following the nomenclature above (*Figure 1*). As observed in previous maps, density associated with a hydrophobic molecule was present in the pore of both maps (*Figure 1—figure supplement 3a*).

For the LMNG-N$_{Const}$ conformation, the density for the side-chains of the residues of the N-terminus and the following link to TM1 is much clearer than seen in the other maps (*Figure 1c*), however it remains difficult to place the first three residues of the N-terminus unambiguously. This new structure is an advance on the previous structure of the equivalent conformation obtained in DDM (PDB 7QEW) (*Figure 1—figure supplement 3c*). In addition to being able to assign more residues to the density in LMNG-N$_{Const}$, there are two main regions which have been modelled differently. The first area that differs is in TM1 (*Figure 1c*, *Figure 1—figure supplement 3b*). Previously we noted variation between the Cx26 crystal structures and our cryo-EM structures in the position of the residues between Val37 and Glu42 (*Brotherton et al., 2022*). In the LMNG-N$_{Const}$ structure, the N-terminal region of TM1, preceding this area and comprising residues Gly21 to Phe31, is rotated with respect to that modelled previously, changing the position of the π-helix in TM1 from residues Ile20-Leu25 to residues Phe29-Val38 (*Figure 1—figure supplement 3b*). Thus, the conformation of TM1 in the LMNG-N$_{Const}$ structure differs from the crystal structures at both the N-terminal and C-terminal ends through variations in the twist of the helical repeats.

The second difference is at the cytoplasmic side of TM3. In all structures of connexins solved to date, the cytoplasmic loop has been disordered. For Cx26 this region extends from approximately residue Arg104 at the C-terminus of TM2 to Glu129 at the N-terminus of TM3. In the crystal structure (Cx26-xtal) (*Maeda et al., 2009*) residues 125–129 have been modelled as part of TM3, whereas in our previous structures solved by cryo-EM (*Brotherton et al., 2022*) there was no evidence in the associated maps of the helix extending beyond Glu129. Instead, when analysing these maps it was noted that unassigned density protruded from near to the top of TM3 towards the loop between the N-terminus and TM1 in a manner that resembles models from AlphaFold2 (*Jumper et al., 2021*; *Varadi et al., 2022*). In the LMNG-N$_{Const}$ map this density is much more clearly defined as the C-terminal end of the cytoplasmic loop as it joins onto TM3 (*Figure 1b*, *Figure 1—figure supplement 2*). Residues Gln124 to Ile128, which form part of the K$_{125}$VRIEG$_{130}$ motif, important for carbamylation, were modelled into it, with Val126 located just above the linking region between the N-terminal helix and TM1 (*Figure 1b*). Though the density for the side-chains is poor and there is no definitive interaction, Lys125 in this position is relatively close to the side-chain of Arg104 of the neighbouring subunit, with which it had been proposed to form a salt bridge following carbamylation (*Figure 1c*).

By contrast, in the LMNG-N$_{Flex}$ map neither the N-terminus nor the KVRIEG motif are well defined (*Figure 1a*). The LMNG-N$_{Flex}$ map is reminiscent of the map derived from Cx26 particles vitrified under low PCO$_2$ conditions (*Brotherton et al., 2022*). In the associated structure the conformation of TM1 is the same as was modelled for the previous structures in DDM. A comparison between the two conformations derived from the new data is shown in *Figure 2a and b* and *Figure 2—videos 1 and 2*. The conformational change in TM1 results in the side-chain of Trp24 rotating by ~90° between the two extreme positions (*Figure 2b*, *Figure 2—video 2*). At one of these extremes, it faces the exterior of the protein, and nestles a detergent or lipid tail. At the other extreme, it is within the core of the protein next to Arg143 and Ala88 (*Figure 2b*). The conformation in the N$_{Flex}$ structure would not be compatible with the position adopted by TM2 in the LMNG-N$_{Const}$ structure because Thr86 and Leu89 would clash with Phe31 and Ile30 on TM1 of the neighbouring subunit (*Figure 2b*, *Figure 2—videos 1 and 2*). The rotation of TM1 changes not only how TM1 interacts with the N-terminus, but also the conformation of the linker between the N-terminal helix and TM1 (*Figure 2a*). This in turn would not be compatible with the conformation of the KVRIEG motif in LMNG-N$_{Const}$ (*Figure 2b*). Overall, therefore the constriction of the channel by the N-terminal helix is associated with changes in the positions of TMs 1, 2 and the KVRIEG motif of the cytoplasmic loop.

**Table 2.** Cryo-EM refinement and validation statistics.

| | K125E$_{90}$ | LMNG$_{90}$ | | | |
|---|---|---|---|---|---|
| | | N$_{Flex}$ | N$_{Const}$ | N$_{Const-mon}$ | N$_{Flex-mon}$ |
| Deposited structure PDB ID | 8Q9Z | 8QA1 | 8QA0 | 8QA2 | 8QA3 |
| Final particle number | 161,625 | 59,005 | 35,007 | 357,859 | 240,137 |
| Map resolution | | | | | |
| FSC threshold | 0.143 | 0.143 | 0.143 | 0.143 | 0.143 |
| Symmetry imposed | C6 | C6 | C6 | C1 | C1 |
| Unmasked (Å) | 2.3 | 2.2 | 2.4 | 2.6 | 2.4 |
| Masked (Å) (masked) | 2.4 | 2.0 | 2.1 | 2.3 | 2.2 |
| Refinement | | | | | |
| Initial model (PDB code) | 7QEQ | 7QEQ | K125E$_{90}$ | AlphaFold2 | N$_{Const-mon}$ |
| Resolution (Å; FSC = 0.5) model | 2.6 | 2.4 | 2.6 | 2.6 | 2.6 |
| Sharpening B factor (Å$^2$) | Local | Local | Local | Local | Local |
| Model composition | 12 chains | 12 chains | 12 chains | 12 chains | 12 chains |
| Non-hydrogen atoms | 19,704 | 18,720 | 19,878 | 18,945 | 18,580 |
| Protein residues | 2316 | 2190 | 2340 | 2260 | 2226 |
| Water | 348 | 318 | 342 | 322 | 254 |
| Ligand: lipid/detergent | 48 | 24 | 24 | 13 | 12 |
| B factor (Å$^2$) | | | | | |
| Protein | 66 | 54 | 77 | 68 | 73 |
| Water | 42 | 33 | 56 | 45 | 44 |
| Lipid/detergent | 56 | 60 | 85 | 69 | 70 |
| R.m.s. deviations | | | | | |
| Bond lengths (Å) | 0.004 | 0.002 | 0.004 | 0.002 | 0.003 |
| Bond angles (°) | 0.516 | 0.484 | 0.593 | 0.450 | 0.544 |
| Validation | | | | | |
| MolProbity score | 1.54 | 1.25 | 1.55 | 1.53 | 1.74 |
| Clashscore | 5.66 | 3.46 | 6.50 | 4.90 | 5.20 |
| Rotamer outliers (%) | 1.83 | 1.42 | 1.05 | 2.32 | 3.71 |
| Ramachandran plot | | | | | |
| Favoured (%) | 97.84 | 98.97 | 96.99 | 98.5 | 97.89 |
| Allowed (%) | 2.16 | 1.03 | 3.01 | 1.49 | 2.11 |
| Disallowed (%) | 0 | 0 | 0 | 0 | 0 |
| CaBlam outliers (%) | 1.53 | 1.24 | 1.6 | 0.83 | 1.08 |
| Correlation coefficients | | | | | |
| CC (mask) | 0.86 | 0.87 | 0.85 | 0.86 | 0.87 |
| CC (box) | 0.7 | 0.71 | 0.66 | 0.68 | 0.70 |
| CC (peaks) | 0.68 | 0.70 | 0.63 | 0.66 | 0.68 |
| CC (volume) | 0.84 | 0.85 | 0.84 | 0.85 | 0.85 |
| Mean CC for ligands | 0.71 | 0.73 | 0.61 | 0.68 | 0.70 |

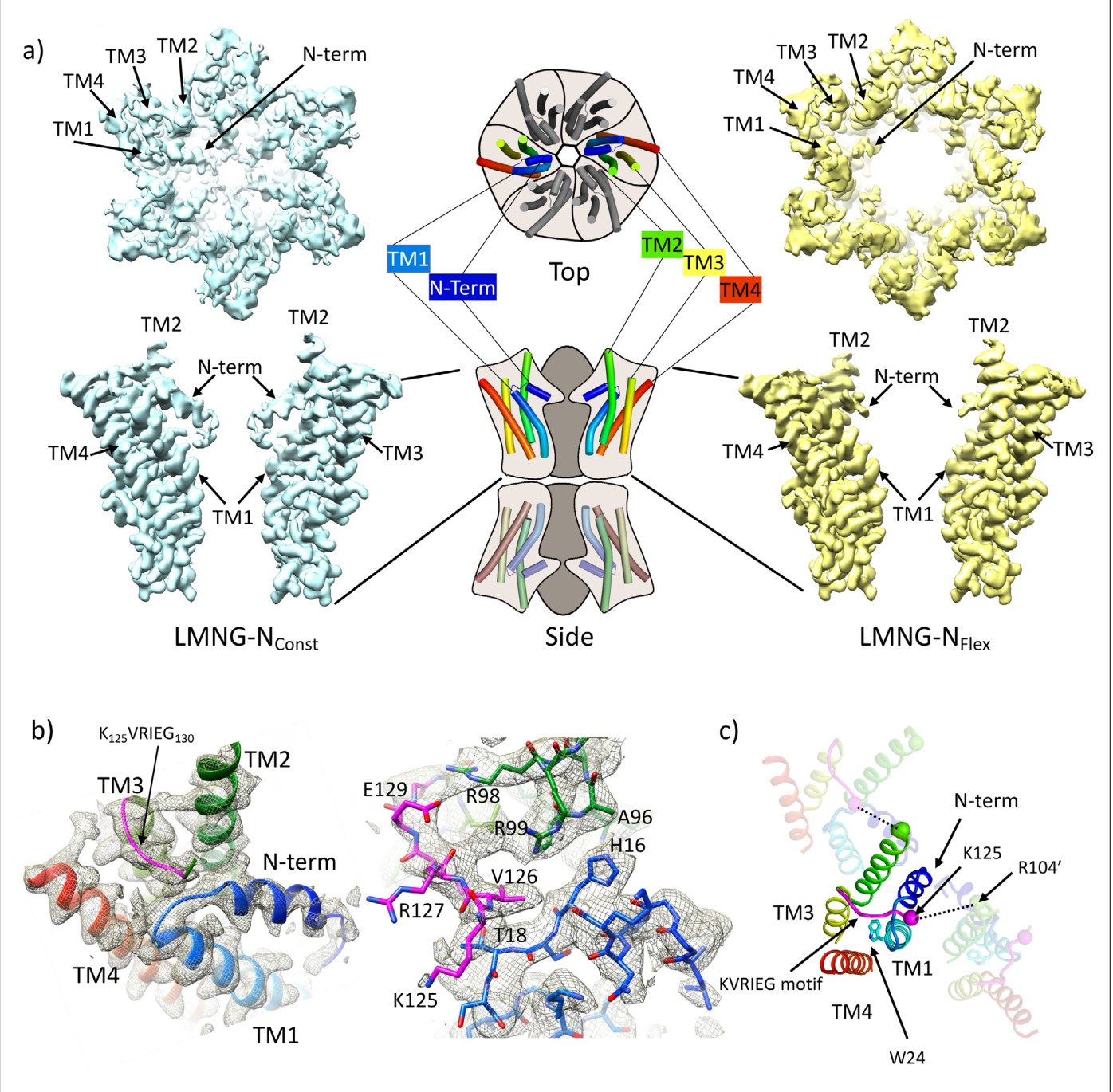

**Figure 1.** Distinct classes from classification of connexin26 (Cx26) solubilised in lauryl maltose neopentyl glycol (LMNG). (**a**) Overall density associated with LMNG-N$_{Const}$ (left) and LMNG-N$_{Flex}$ (right) viewed from the cytoplasmic face (top) and from the side (bottom; for clarity only two subunits are shown). To orientate the reader a schematic of the full connexin gap junction channel (GJC) is shown in the centre. The cartoon showing the N-terminal helix and the transmembrane helices (TMs) has been coloured through the colours of the rainbow with blue at the N-terminus to red at the C-terminus. (**b**) As (**a**) focussed on the KVRIEG motif and the link between the N-terminus and TM1. Left: The cartoon has been coloured as in (**a**) except for the KVRIEG motif, which is shown in magenta. Right: Stick representation with the same colouring showing the interactions between the residues on the link between the N-terminus and TM1 (blue), residues on TM2 (green) and the KVRIEG motif (magenta). (**c**) Cartoon representation of the cytoplasmic region of the LMNG-N$_{Const}$ structure. The two neighbouring subunits to the central subunit in the figure have been made semi-transparent. The dotted lines show the proximity of K125 of one subunit to R104 of the neighbouring subunit. Trp24 on TM1 is in the region of TM1 that adopts an altered conformation with respect to the previously solved structures of Cx26.

The online version of this article includes the following video and figure supplement(s) for figure 1:

*Figure 1 continued on next page*

*Figure 1 continued*

**Figure supplement 1.** Workflow for initial processing of cryo-EM data for wild-type (WT) sample, solubilised in lauryl maltose neopentyl glycol (LMNG) in $CO_2$/$HCO_3^-$ buffer.

**Figure supplement 2.** Density for the transmembrane helices and the N-terminal helix associated with each the lauryl maltose neopentyl glycol (LMNG)-$N_{Const}$ and LMNG-$N_{Flex}$ structures.

**Figure supplement 3.** Comparison of structures derived from lauryl maltose neopentyl glycol (LMNG) solubilised protein with the structure derived from the dodecyl β-D-maltoside (DDM) solubilised protein.

**Figure 1—video 1.** Morph showing the conformational differences between reconstructions of lauryl maltose neopentyl glycol (LMNG)-$N_{Const}$ and LMNG-$N_{Flex}$.

https://elifesciences.org/articles/93686/figures#fig1video1

## Density for cytoplasmic loop compatible with models from AlphaFold2

Despite the LMNG-$N_{Const}$ map being much clearer for the cytoplasmic region of the protein, residues from 107 to 123 were still missing. We, therefore, carried out another classification of the particles focussed on the cytoplasmic region of a single subunit (see Methods). As above, this resulted in a range of maps showing varying positions of the TMs and clarity of the N-terminal helix (*Figure 3—figure supplement 1*). Importantly, in one case and where the N-terminus was clearly defined, extra density was also seen for the cytoplasmic loop, albeit at low resolution. The structure was tentatively

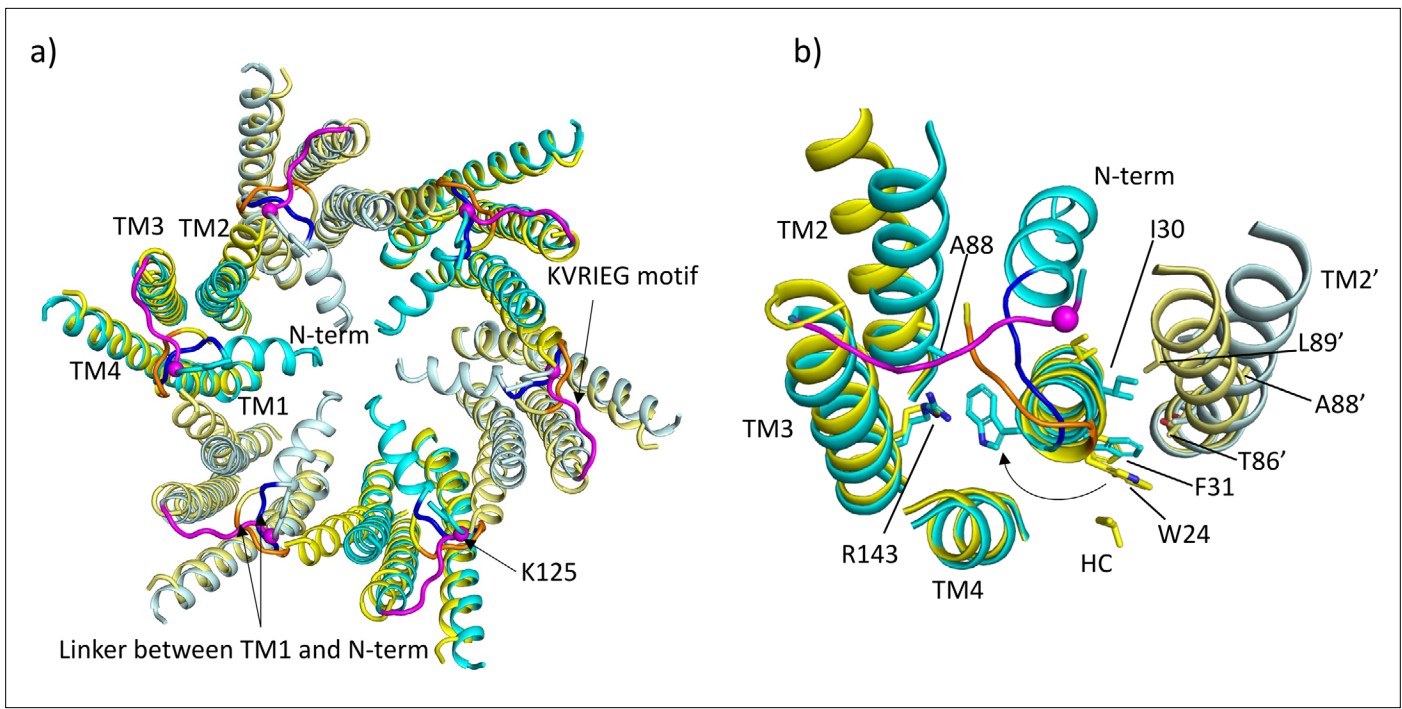

**Figure 2.** Comparison of lauryl maltose neopentyl glycol (LMNG)-$N_{Const}$ and LMNG-$N_{Flex}$ structures. (**a**) Overall superposition showing the movement of TM2 and the link between the N-terminus and TM1. LMNG-$N_{Const}$ in cyan and LMNG-$N_{Flex}$ in yellow (alternate subunits have been coloured in lighter shades). The KVRIEG motif has been coloured magenta with a sphere indicating the position of K125. The residues between the N-terminus and TM1 for the LMNG-$N_{Const}$ structure have been coloured blue. (**b**) As (**a**) but focussed on TM1. The conformation of TM1 differs between the two structures as seen by the change in position of Trp24. TM2′ is from the neighbouring subunit. HC denotes the hydrocarbon chain from a lipid. The positions of T86′ and L89′ of TM2 in the $N_{Flex}$ conformation are not compatible with F31 and I30 TM1 in the $N_{Const}$ conformation.

The online version of this article includes the following video(s) for figure 2:

**Figure 2—video 1.** Morph showing the conformational differences between structures of lauryl maltose neopentyl glycol (LMNG)-$N_{Const}$ and LMNG-$N_{Flex}$.

https://elifesciences.org/articles/93686/figures#fig2video1

**Figure 2—video 2.** Morph showing the conformational differences between structures of lauryl maltose neopentyl glycol (LMNG)-$N_{Const}$ and LMNG-$N_{Flex}$.

https://elifesciences.org/articles/93686/figures#fig2video2

built into the density with the cytoplasmic loop of the classified subunit in a conformation resembling models from AlphaFold2 (*Jumper et al., 2021*; *Figure 3*) and with a complete N-terminus (*Table 2*, *Figure 3—figure supplement 2*). Only residues 109–114 were omitted in the final structure as the placement of these residues was ambiguous. In the maps associated with this structure there is density that we cannot assign, near to Lys125, between Ser19 in the TM1-N-term linker, Tyr212 of TM4, and Tyr97 on TM3 of the neighbouring subunit, which may be a small molecule that has bound. Overall, the conformation of the subunit is very similar to the LMNG-$N_{Const}$ structure from the classification based on the masked hemichannels, with an RMSD of 0.38 Å for 198 Cα atoms. A second structure was also refined that had a conformation much more similar to the LMNG-$N_{Flex}$ structure (LMNG-$N_{Flex-mon}$) (RMSD 0.45 for 176 Cα atoms) (*Figure 3—figure supplements 1 and 2*, *Table 2*). Rather surprisingly, given the success of the hemichannel mask-based classification, only the subunit upon which the focussed classification had been carried out had this conformation, with the density from the other subunits appearing more like the map before focussed classification. When hexameric symmetry was applied to the subunit, though the conformation of the N-terminus caused the aperture of the pore to appear closed, steric clashes involving the N-terminal residues suggested a symmetric arrangement of this conformation would not be possible (*Figure 3d*). This analysis suggested a picture of a flexible molecule that can be captured in different conformations ranging from closed to open but with limited cooperativity between the subunits of the hexamer.

## Mutation of K125 to glutamic acid results in constitutively closed GJCs

The above data clearly showed two conformations, from which we could infer a mechanism for closure of the pore. We had previously shown that in GJCs, the K125R mutation remains in the open state even under conditions of high $PCO_2$ (*Nijjar et al., 2021*). Based on our experience with mutations of Cx26 we hypothesised that, if K125E results in constitutively open hemichannels, the same mutation would result in constitutively closed GJCs. Thus, if it were to be true, we could investigate the structure of the proteins under identical buffer conditions where the channel was biased towards open or closed conformations. To verify the effect of mutating Lys125 to a glutamic acid, we used an established dye transfer assay between coupled cells to assess gap junction function (*Nijjar et al., 2021*). For wild-type (WT) Cx26, gap junctions readily form between cells and allow rapid transfer of dye from a donor (filled via a patch pipette) to a coupled acceptor cell at a $PCO_2$ of 35 mmHg (*Figure 4a*). Cx26[K125E] forms structures that resemble WT gap junctions (*Figure 4b*). However, these gap junctions appeared to be shut and did not permit dye transfer at a $PCO_2$ of 35 mmHg (*Figure 4b and d*). As the action of an increase in $PCO_2$ is to close the WT Cx26 gap junction, unsurprisingly Cx26[K125E] gap junctions remained closed at a $PCO_2$ of 55 mmHg (*Figure 4c and d*).

## The K125E mutation biases the conformational equilibrium to the $N_{Const}$ structure

Given that the K125E mutant resulted in constitutively closed channels and the K125R mutant in channels that do not close in response to $CO_2$, we set out to solve the respective structures. With respect to the WT and K125R constructs, purification of the K125E protein resulted in higher yields, consistent with a more stable protein. For both proteins cryo-EM data were collected from protein solubilised in DDM and vitrified in $CO_2$/$HCO_3^-$ buffers corresponding to a $PCO_2$ of 90 mmHg with the pH maintained at pH 7.4 as was done previously (*Brotherton et al., 2022*). Refinement with D6 symmetry imposed resulted in maps with nominal resolutions of 2.2 and 2.1 Å respectively as defined by gold standard FSC (*Rosenthal and Henderson, 2003*; *Scheres, 2012*; *Table 1*, *Figure 5—figure supplements 1 and 2*). Superposition of the two maps showed there was a small but distinct change in the position of the cytoplasmic portion of TM2 between the two D6 averaged maps (*Figure 5a and b*, *Figure 5—video 1*). Of the two, the K125R map looked much more similar to the equivalent map from the WT protein purified in the same way in DDM (PDB ID 7QEQ) and vitrified at the same $PCO_2$ (*Figure 5c*). Further classification focussed on the cytoplasmic region of one hemichannel of the GJCs provided further evidence of a distinct difference in the conformations of the proteins. For the K125E data set the most populated class (43% of the particles) had a conformation similar to the LMNG-$N_{Const}$ (*Figure 5—figure supplement 1*). In contrast only 10% of the data for the K125R belonged to this class (*Figure 5—figure supplement 3*).

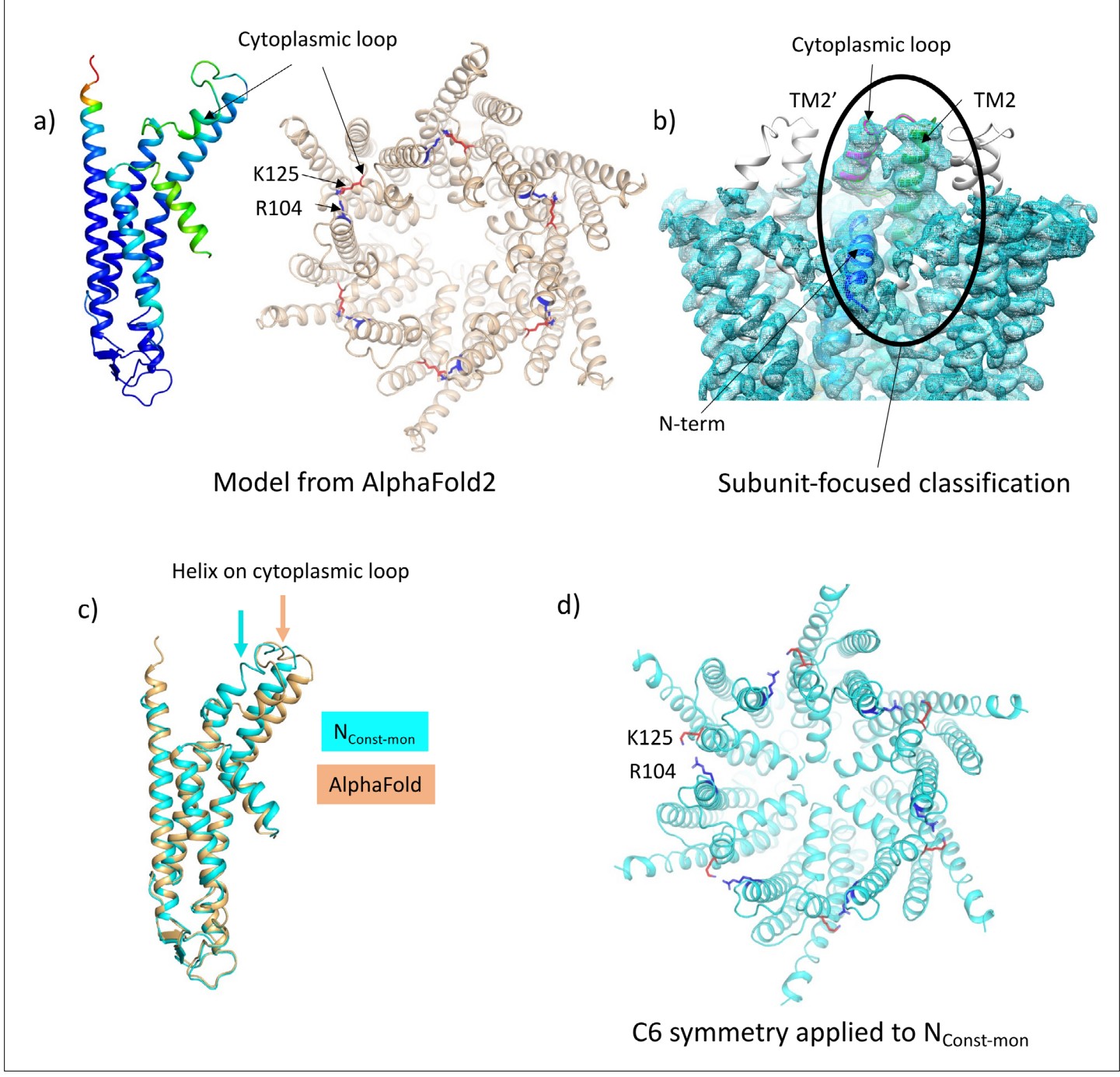

**Figure 3.** Focussed classification of a single subunit results in density for the cytoplasmic loop consistent with models from AlphaFold2. (**a**) Models generated by AlphaFold2 for a single subunit (left; coloured according to confidence level) and for the hexamer (right; in wheat with the position of K125 depicted by red sticks and the position of R104 in blue). (**b**) Focussed classification of a single subunit (highlighted by an oval and coloured as in *Figure 1c* with the cytoplasmic loop in magenta) resulted in clear density for part of the cytoplasmic loop in a conformation consistent with the models from AlphaFold2. This does not extend to the neighbouring subunits (the conformation of the subunit is replicated in grey for the neighbouring subunits). (**c**) Superposition of the single subunit built into the density (cyan) on the AlphaFold2 model (wheat). Showing the change in position of the helix in the cytoplasmic loop (highlighted by an arrow in the relevant colour). (**d**) Reconstituting a hexamer by replicating the conformation of the subunit seen in (**b**) to all six subunits of the hexamer results in an apparently more closed conformation of the hemichannel, though there are also residue clashes, especially at the N-terminus. Lys 125 and Arg 104 are depicted with red and blue sticks, respectively.

The online version of this article includes the following figure supplement(s) for figure 3:

**Figure supplement 1.** Workflow for single subunit classification for lauryl maltose neopentyl glycol (LMNG) solubilised sample.

*Figure 3 continued on next page*

*Figure 3 continued*

**Figure supplement 2.** Density for the transmembrane helices and the N-terminal helix associated with each the lauryl maltose neopentyl glycol (LMNG)-$N_{Const-mon}$ and LMNG-$N_{Flex-mon}$ structures.

Reconstructions from the most predominant class from the K125E data set ($K125E_{90}$) have a nominal resolution of 2.5 Å (*Table 2*, *Figure 5—figure supplement 1*). The KVRIEG motif and the N-terminus are reasonably well defined in the density as seen for the LMNG-$N_{Const}$ structure (*Figure 5d*) though again, the first three residues are difficult to position. Overall, the conformation is very similar to LMNG-$N_{Const}$ with an RMSD of 0.35 Å for 195 out of 199 matched Cα pairs (*Figure 5e*). In contrast the RMSD compared to the LMNG-$N_{Flex}$ conformation is 2 Å across 180 Cα pairs.

The difference of the K125E maps from the equivalent maps from the WT and K125R proteins indicated a clear bias towards a more closed conformation of the protein in the $CO_2/HCO_3^-$ buffers. To understand the contribution of the K125E mutant, independently of any effect of $CO_2$ we also reanalysed data collected from both WT and K125E protein vitrified in HEPES buffer at pH 7.4. While the resolution of the maps was much lower for these reconstructions (*Table 1*, *Figure 5—figure supplement 4*) and better for the K125E mutant (4.2 Å) than the WT protein (4.9 Å) superposition of the respective maps again showed a movement of the cytoplasmic portion of TM2 together with a change in the N-terminus (*Figure 5—figure supplement 5* and *Figure 5—video 1*). Thus, it appears that the K125E mutant is sufficient by itself to bias the conformation in the absence of $CO_2$. However, $CO_2$ may have other effects on the protein to give the higher resolution and more defined conformation seen in the $CO_2/HCO_3^-$ buffers.

## Discussion

Connexins open and close to various stimuli. Despite a number of recent high-resolution structural studies of connexins, the structural basis for pore closure or opening in response to a stimulus is still poorly understood. Cx26 is affected by $PCO_2$. We have previously shown that there is a correlation in the closure of the Cx26 GJC with the level of $PCO_2$ and also with the position of TM2 (*Brotherton et al., 2022*). Our improved data show that, not only does pore closure and the position of the N-terminal helix correlate with TM2, but also with the conformation of TM1 and importantly, the KVRIEG motif on the regulatory cytoplasmic loop.

The conformation of the KVRIEG motif and the cytoplasmic loop is interesting. In the original low-resolution crystal structure of Cx26 (Cx26-xtal) (*Maeda et al., 2009*), the KVRIEG motif is modelled as part of TM3. However, in most other structures of connexin GJCs, the equivalent residues are not observed in the density, and it appears there is a breakdown in helical conformation at this point. In the $N_{Const}$ structures, the KVRIEG motif forms an extended conformation mimicking the conformations seen in AlphaFold2 structures. In fact, the conformation of the complete loop that we observe in the subunit focussed classification is similar to that seen in AlphaFold2 structures (*Varadi et al., 2022*). Essentially the KVRIEG motif can be described as a break in TM3 as the helix extends at either side of this extended motif. Although the low resolution of the density of the complete loop prevents the accurate positioning of the residues, the combination of our structure and the predicted models paves the way for further studies of the importance of particular interactions in the loop.

From a comparison of our structures with those of other connexins it is apparent that the position of TM2 seen in the $N_{Const}$ structures is an outlier, with a rotation of the cytoplasmic region of the helix compared to other structures (*Figure 6a*). With respect to the N-terminus the various reported structures can be considered in three main categories: those with the N-terminus folded into the channel pore (N-down) (*Flores et al., 2020*; *Lee et al., 2023a*; *Lee et al., 2023b*; *Myers et al., 2018*); those where the N-terminus is lifted (N-up) (*Lee et al., 2020*; *Qi et al., 2023a*); and those where the N-terminus is flexible (N-flex) (*Bennett et al., 2016*; *Brotherton et al., 2022*; *Khan et al., 2020*; *Lee et al., 2023b*; *Qi et al., 2023b*). The conformation of the Cx26-$N_{Const}$ structure most resembles the structures with N-down as these have similar conformations of TM1 (*Figure 6*). However, whereas the N-terminus in these latter structures folds flat against the pore of the channel to give an open aperture, in the Cx26-$N_{Const}$ structure, it would be prevented from doing so by the position of TM2 and so is forced outwards to form a much more constricted aperture. The conformation of TM1 in the Cx26-$N_{Flex}$ structures, which we consider to be open, is more similar to structures of other connexins in the N-flex or

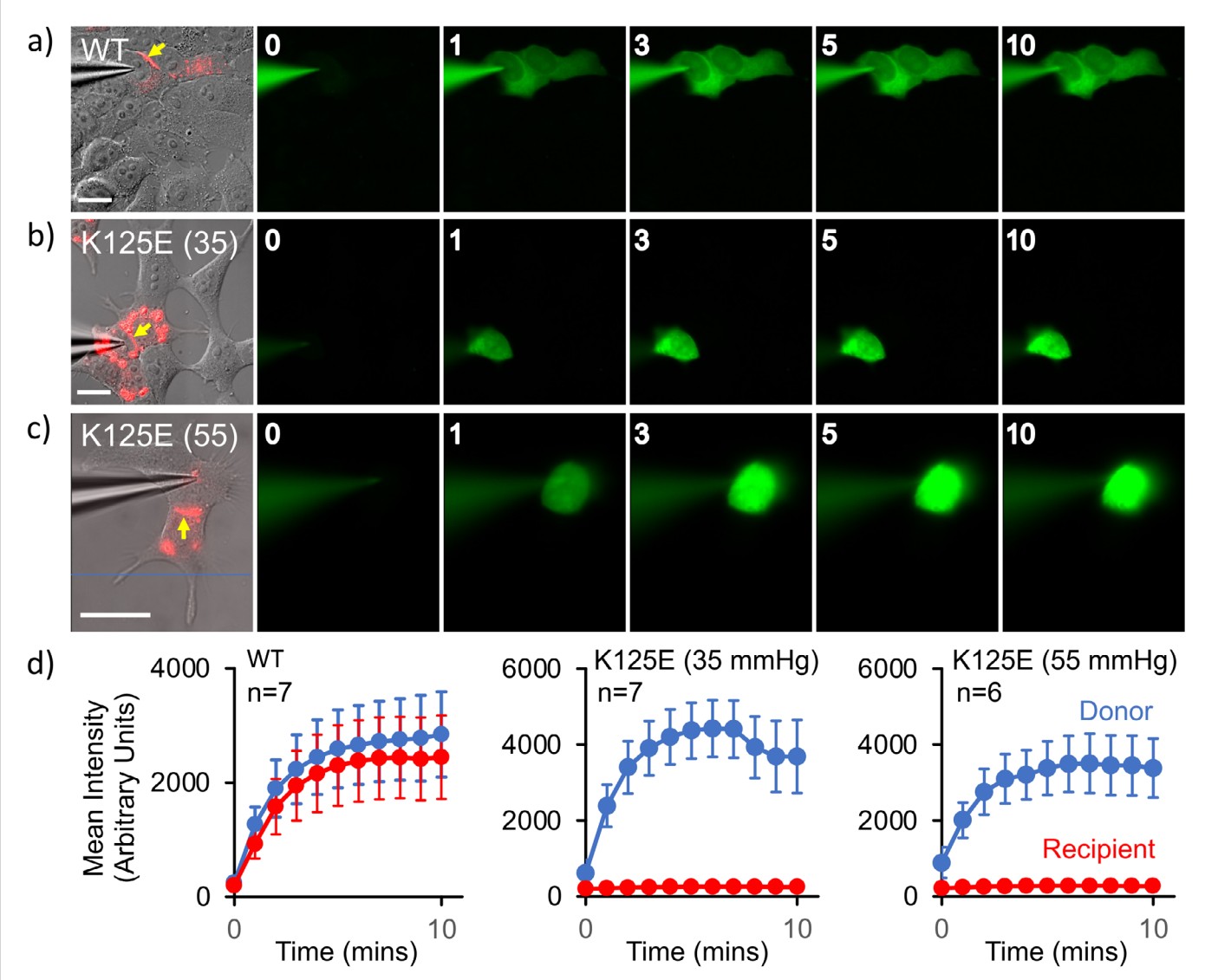

**Figure 4.** Cx26$^{K125E}$ gap junctions are constitutively closed at a partial pressure of CO$_2$ (PCO$_2$) of 35 mmHg. (**a–c**) Montages each showing bright-field DIC image of HeLa cells with mCherry fluorescence corresponding to the Cx26$^{K125E}$-mCherry fusion superimposed (leftmost image) and the permeation of 2-Deoxy-2-[(7-nitro-2,1,3-benzoxadiazol-4-yl)amino]-D-glucose (NBDG) from the recorded cell to coupled cells. Yellow arrow indicates the presence of a gap junction between the cells; scale bars, 20 μm. The numbers are the time in minutes following the establishment of the whole-cell recording. In Cx26$^{WT}$ expressing cells (**a**), dye rapidly permeates into the coupled cell. For Cx26$^{K125E}$ expressing cells, no dye permeates into the neighbouring cell even after 10 min of recording at either 35 (**b**) or 55 mmHg (**c**) PCO$_2$ despite the presence of morphological gap junctions. (**d**) Quantification of fluorescence intensity in the recorded cell (donor) and the potentially coupled cell (recipient) for both Cx26$^{WT}$ and Cx26$^{K125E}$ (seven pairs of cells recorded for WT and K125E at 35 mmHg and six pairs of cells for K125E at 55 mmHg, data presented as mean ± SEM). While dye permeation to the recipient cell follows the entry of dye into the donor for Cx26$^{WT}$, no dye permeates to the acceptor cell for Cx26$^{K125E}$. Note that the fluorescence intensity in the donor cell for Cx26$^{K125E}$ at both levels of PCO$_2$ is higher than for Cx26$^{WT}$ at 35 mmHg, presumably because the dye remains trapped in the donor cell rather than diffusing to the recipient cell.

The online version of this article includes the following source data for figure 4:

**Source data 1.** Quantification of fluorescence intensity in the recorded cell (donor) and the potentially coupled cell (recipient) for both Cx26$^{WT}$ and Cx26$^{K125E}$.

N-up conformations. For Cx36 and Cx43 these conformations have been suggested to represent the closed conformation (*Lee et al., 2023a*; *Qi et al., 2023a*), though for Cx32 this has been described as an open conformation (*Qi et al., 2023b*). The apparent outlier amongst the Cx26 structures is the Cx26-xtal structure where the N-terminus points into the pore, but the overall conformation is more

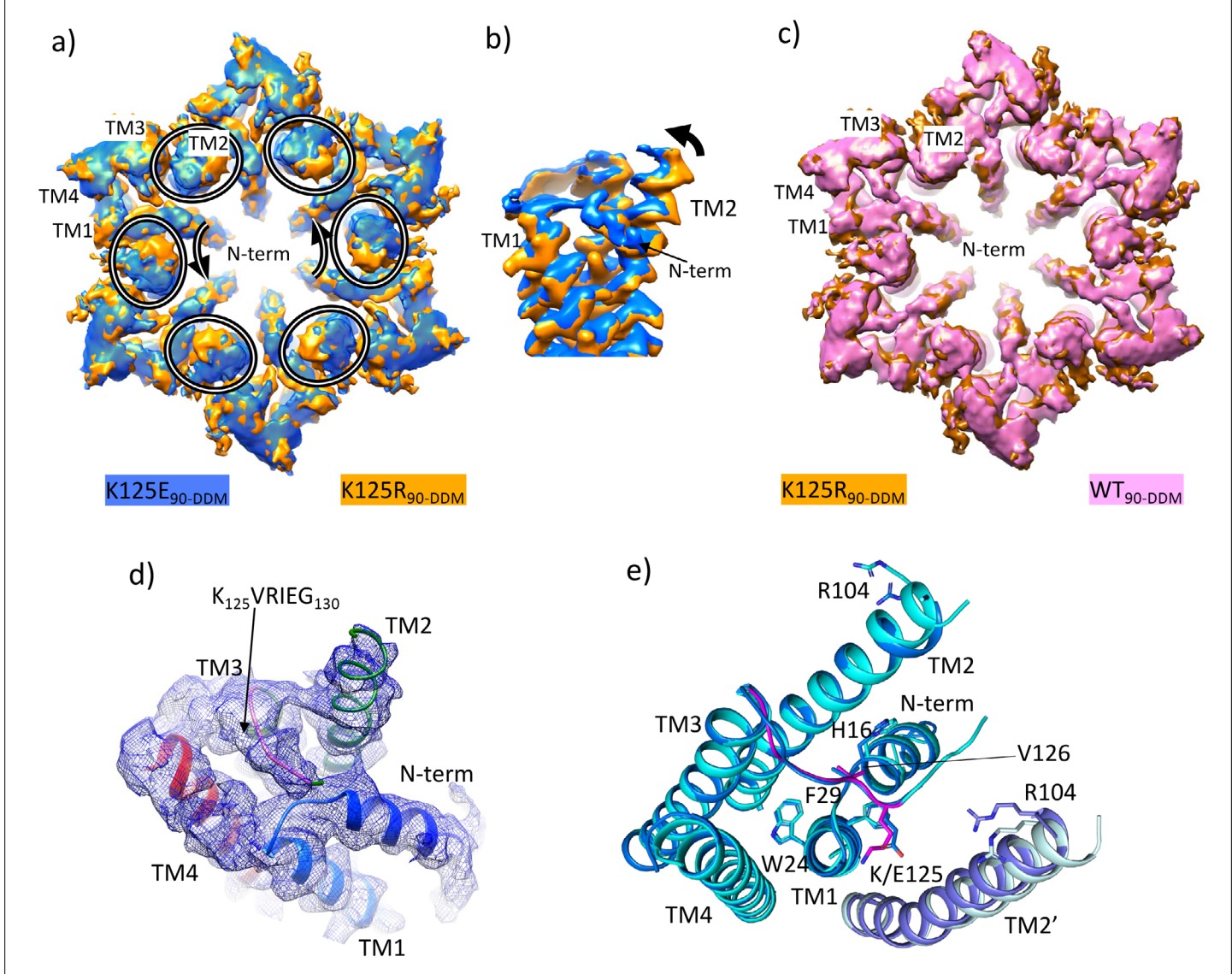

**Figure 5.** Density associated with the K125E mutant. (**a**) Superposition of density for K125E$_{90}$ D6 averaged map (blue) on the density for the K125R$_{90}$ D6 averaged maps (orange). The ovals show the position of TM2 from each subunit and the arrows show the direction of the difference between TM2 in the two structures. (**b**) As (**a**) but focussed on TM2 in a view approximately perpendicular to the membrane. (**c**) Superposition of density for WT$_{90}$ connexin26 (Cx26) (PDB ID 7QEQ; pink) D6 averaged maps on the density for R125E$_{90}$ (orange). (**d**) Density associated with one subunit of the K125E$_{90}$ structure (unsharpened map). The structure has been coloured as in *Figure 1c*. (**e**) Superposition of K125E$_{90}$ structure (light blue) on the structure of lauryl maltose neopentyl glycol (LMNG)-N$_{Const}$ (cyan) showing the similarity between the two structures.

The online version of this article includes the following video and figure supplement(s) for figure 5:

**Figure supplement 1.** Workflow for processing of cryo-EM data for K125E sample in $CO_2$/$HCO3^-$ buffer.

**Figure supplement 2.** Density for the transmembrane and N-terminal helix associated with the K125E structure.

**Figure supplement 3.** Workflow for processing of cryo-EM data for K125R sample in $CO_2$/$HCO3^-$ buffer.

**Figure supplement 4.** Workflows for processing of cryo-EM data for samples in HEPES buffer.

**Figure supplement 5.** Comparison of density maps from wild-type (WT) and K125E connexin26 (Cx26) purified in HEPES buffer at pH 7.4.

**Figure 5—video 1.** Morph showing the conformational difference between D6 refined reconstructions of K125E and K125R.
https://elifesciences.org/articles/93686/figures#fig5video1

**Figure 5—video 2.** Morph showing the conformational differences between D6 refined reconstructions of wild-type (WT) and K125E in HEPES buffer.
https://elifesciences.org/articles/93686/figures#fig5video2

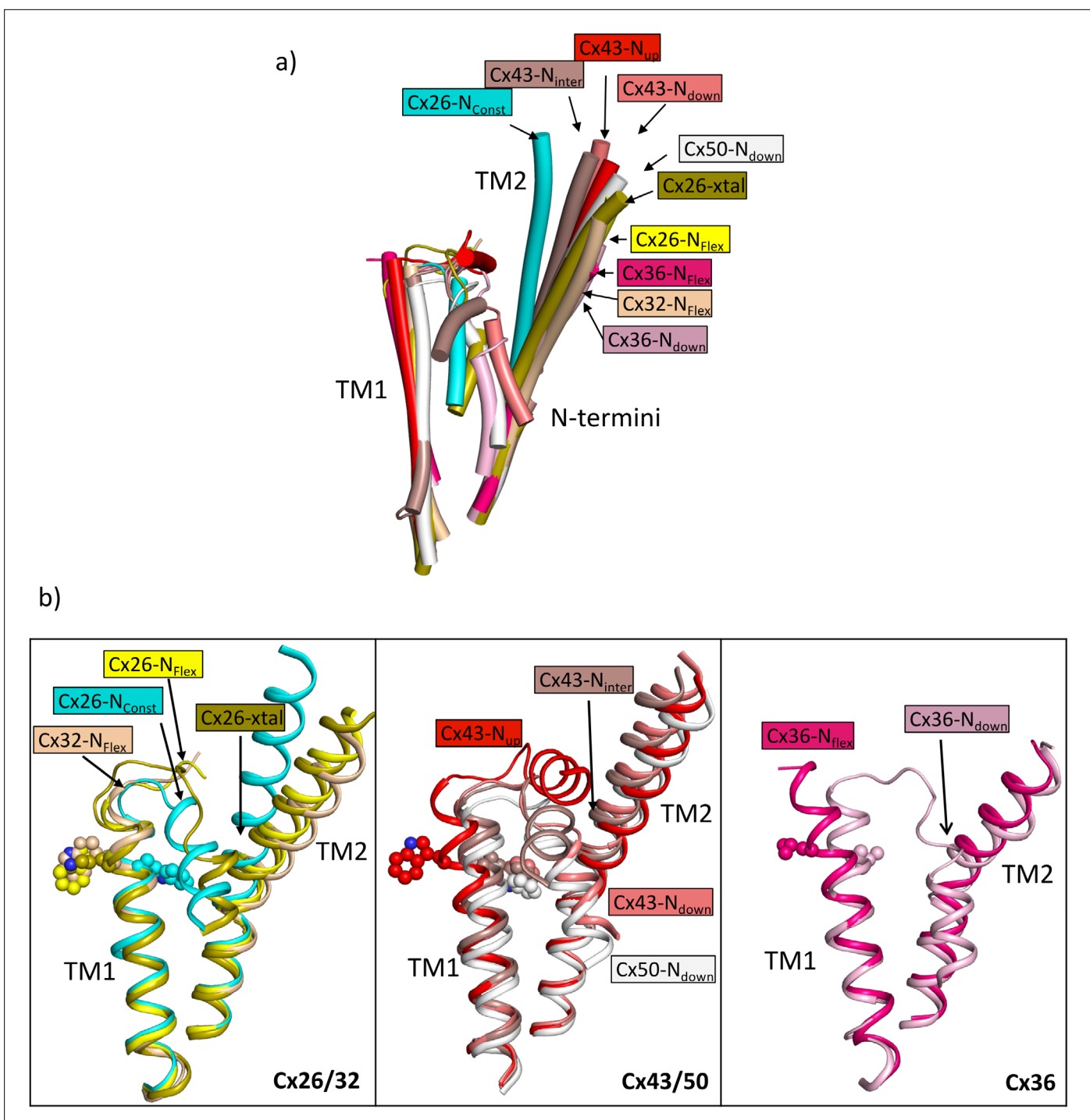

**Figure 6.** Comparison of the two structures derived from the lauryl maltose neopentyl glycol (LMNG) classification with other structures of connexins. (**a**) Superposition of a single subunit from the LMNG-$N_{Const}$ (cyan) and LMNG-$N_{Flex}$ (yellow) structures on: connexin26 (Cx26) crystal structure (chartreuse, PDB ID 2ZW3); Cx32 (wheat, 7zxm) Cx50 (white, 7JJP); Cx43 in up (red, 7XQF), intermediate (chocolate, 7XQI), and down (salmon, 7F94) conformations; Cx36 in down (pink, 7XNH) and flexible (raspberry, 7XKT) conformations. The structures were superposed based on all chains of the hexamer. For clarity, only TM1, TM2, and the N-terminal helices are shown for each structure. (**b**) As (**a**) for the beta connexins Cx26 and Cx32 (left), alpha connexins Cx43 and Cx50 (middle), and the gamma connexin Cx36 (right) structures separately. Trp24 in each of the Cx26 and Cx43 structures has been depicted with a sphere representation. The isoleucine in the corresponding position is shown for Cx36. The sequence identities for common residues to Cx26 are 63% for Cx32, 49% for Cx50, 43% for Cx43, and 35% for Cx36.

similar to the Cx26-N$_{Flex}$ conformation. However, relative to this latter structure the helix is pulled back (*Figure 6b*) so is more open than seen in the Cx26-N$_{Const}$ structure.

Previously, when examining our Cx26 maps from DDM solubilised protein we noted a lipid-like molecule in the pore of the protein and we questioned whether this would be detergent or lipid and whether it would interfere with the position of the N-terminal helix (*Brotherton et al., 2022*). The fact that similar density remains when the protein is solubilised with LMNG rather than DDM strongly suggests that this molecule cannot be detergent and is more likely to be a lipid. Though the presence of the molecule may still be an artefact of either the solubilisation process or heterologous expression in insect cells, it is interesting to note that a lipid-like molecule in this position appears to be a constant feature of all the high-resolution cryo-EM maps associated with the structures of connexins where the N-terminus is not in the N-down position. This includes both GJCs (*Lee et al., 2023a*; *Lee et al., 2023b*; *Qi et al., 2023a*; *Qi et al., 2023b*) and hemichannels (*Lee et al., 2020*; *Qi et al., 2023b*) and is irrespective of the solubilisation method or whether the structure has been solved in the presence of detergent or nanodiscs. It seems that the lipid is only displaced when the N-terminus adopts the pore lining position. As this position is not possible in the Cx26-N$_{Const}$ structures due to the presence of TM2, the lipid remains.

It must be asked whether the N$_{Const}$ structures that we observe represent the closed state. It is difficult to model the first three residues of the N-terminus unambiguously, presumably due to the breakdown in sixfold symmetry at this point, but with minor changes to the side-chains, the centre of the pore could be closed. The presence of the lipid would seal any other apertures between the neighbouring N-termini. Rather surprisingly given that we can isolate a symmetrical conformation of the protein using C6 symmetry, reconstructions of the full dodecamer following the focussed classifications of the single subunit do not show an influence of the conformation of that subunit on its neighbours. While this is consistent with structural studies on Cx43 gap junctions (*Lee et al., 2023a*) the stochastic nature of the subunit conformations contrasts with studies of Cx26 hemichannels in cell membranes where significant cooperativity has been observed (*de Wolf et al., 2017*).

There is a clear correlation between the results of the dye transfer assays and the structural results, supporting the interpretation that the N$_{Const}$ conformation is more like the closed state. The effect on the structure of Cx26 in changing K125 to a glutamate both in $CO_2$/$HCO_3^-$ and in HEPES buffer is consistent with the hypothesis that the $CO_2$-mediated closure of the gap junctions is caused by the carbamylation of K125. It would also suggest that even under high $PCO_2$ conditions the WT protein is not completely carbamylated during vitrification. Given that the carbamylation reaction is highly labile, this is perhaps not surprising. The hypothesis has been that the negative charge introduced onto the lysine side-chain would enable it to form a salt bridge with Arg104 from the neighbouring subunit (*Meigh et al., 2013*). Consistent with this, mutation of Arg104 to alanine abolishes $CO_2$ sensitivity in both GJCs (*Nijjar et al., 2021*) and hemichannels (*Meigh et al., 2013*), similar to the mutation of Lys125. In models from AlphaFold2, which differ in conformation from the structure upon which the hypothesis was based, it is also notable that the lysine is located next to the arginine (*Figure 3a*). In the N$_{Const}$ structures the position of the Lys/Glu125 main chain is near to Arg104, even though TM2 moves further from TM3 in moving from the N$_{Flex}$ to the N$_{Const}$ conformations. Though there is no clear interaction between the two residues, with minor adjustments of the side-chains, the residues could be made to interact and the absence of a definite interaction may be due to distortions during the vitrification process or radiation damage to which acidic residues are more prone. Glutamate is also much shorter than the carbamylated lysine, so that while the charges would be equivalent, the two residues might not be able to make the same specific interactions. Our results here show that the negative charge at residue 125 is important for altering conformation. For hemichannels, a direct interaction between the two residues (*Meigh et al., 2015*) is strongly supported by a mutant in which both R104 and K125 are replaced by cysteines, allowing a potential cross-link through a disulphide bond. This mutant responds to a change in redox potential in a similar way to which the WT protein responds to $CO_2$ (*Meigh et al., 2015*). However, although both Lys125 and Arg104 are necessary for $CO_2$-dependent gap junction closure, attributing the induction of the conformational change in the gap junction to a single salt bridge between Lys125 and Arg104 may be an over-simplification. For example, there may be other interactions involved and it is possible that multiple carbamylation events contribute to gap junction sensitivity (*Brotherton et al., 2020*). Given that we observe some protein in the N$_{Const}$ conformation even when Arg125 is mutated to

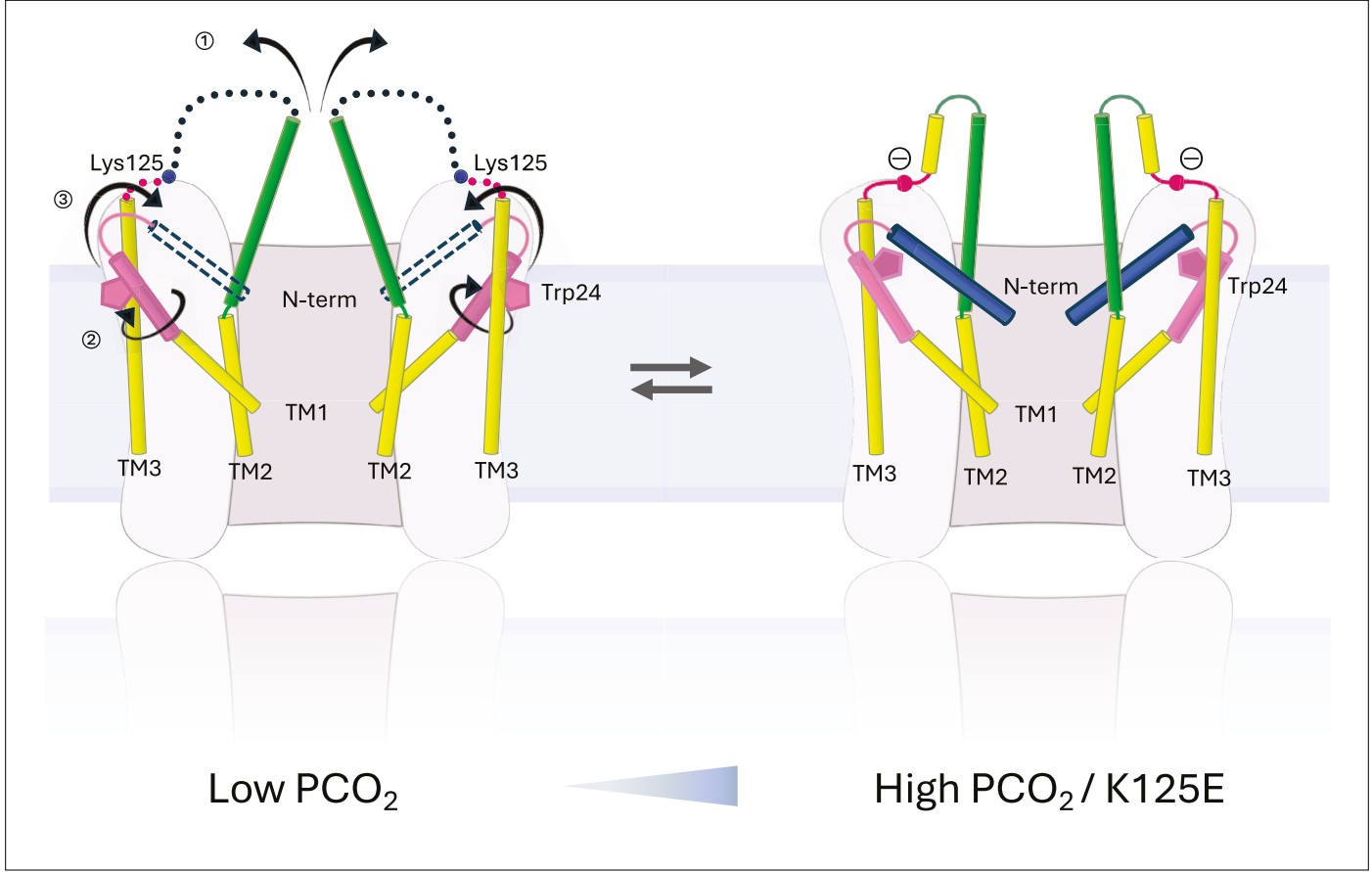

**Figure 7.** Schematic representation of conformational changes. Schematic view of the cytoplasmic region of two opposing subunits within one hemichannel of the gap junction. The open structure on the left and the constricted structure on the right are in a dynamic equilibrium. Increase in partial pressure of $CO_2$ ($PCO_2$) and the introduction of a negative charge on Lys125 of the KVRIEG motif (magenta) pushes the equilibrium to the right. In going from one conformation to the other: ① the cytoplasmic region of TM2 (green) flexes around Phe83 and the cytoplasmic loop adopts a more defined conformation; ② the cytoplasmic region of TM1 (pink) rotates, illustrated here by the movement of Trp24; ③ the N-terminal helix (blue), which will be affected by both ① and ②, adopts a position within the pore that constricts the entrance to the channel.

arginine, our data would be consistent with a conformationally flexible protein, in which the introduction of a negative charge would stabilise one particular conformation, rather than causing a conformational change per se (*Figure 7*).

Mutations in Cx26 lead to both syndromic and non-syndromic deafness (*Xu and Nicholson, 2013*). While these mutations have been mapped on the structure previously, the position of Ala88, mutation of which to Val causes KIDS, is interesting with regard to the new conformations of TM1. This mutation, which is at the flexion point of TM2 (*Figure 2b*), results in leaky hemichannels (*Mhaske et al., 2013*) and has been shown to prevent $CO_2$ sensitivity either to the gap junctions, in closing the channels (*Nijjar et al., 2021*), or to the hemichannels in opening the protein (*Meigh et al., 2014*). In the $N_{const}$ structures the $C_{\beta}$ of Ala88 lies within 4.2 Å of Trp24, which moves during the conformational change of TM1. Replacement of the alanine with the bulkier valine would hinder this conformation from being adopted and therefore, would disfavour the closed conformation. Interestingly, the alanine and the tryptophan are located next to Arg143. Mutation of Arg143 to tryptophan is a very common mutation that leads to non-syndromic hearing loss (*Xu and Nicholson, 2013*). Overall our data, summarised in *Figure 7*, provide important mechanistic insight into the conformational changes behind pore closure in Cx26, which is useful for understanding how mutations in the protein can lead to disease.

# Methods

### Key resources table

| Reagent type (species) or resource | Designation | Source or reference | Identifiers | Additional information |
|---|---|---|---|---|
| Gene (*Homo sapiens*) | GBJ2 | Uniprot | P29033 | |
| Recombinant DNA reagent | pFastbac-human connexin 26 | Gift from Prof. Tomitake Tsukihara and Prof. A Atsunori | | |
| Cell line (*Spodoptera frugiperda*) | Sf9 | Fisher Scientific | CAT# 10500343 | |
| Recombinant DNA reagent | Cx26$^{WT}$ | *Meigh et al., 2013* | | |
| Recombinant DNA reagent | Cx26$^{K125E}$ | *Meigh et al., 2013* | | |
| Cell line (*Homo sapiens*) | HeLa DH (ECACC) | ECACC | CAT# 96112022 RRID:CVCL_2483 | |
| Chemical compound, drug | Insect Xpress | Scientific Laboratory Supplies/Lonza | CAT# LZBELN12-730Q | |
| Chemical compound, drug | DMEM | Merck Life Sciences UK Ltd | CAT# D6046 | |
| Chemical compound, drug | Fetal bovine serum | Labtech.com | CAT# FCS-SA | |
| Chemical compound, drug | GeneJuice Transfection Reagent | Merck Life Sciences UK Ltd | CAT# 70967-3 | |
| Chemical compound, drug | 2-Deoxy-2-[(7-nitro-2,1,3-benzoxadiazol-4-yl)amino]-D-glucose | AAT Bioquest | CAT# 36702 | |
| Chemical compound, drug | K-gluconate | Merck Life Sciences UK Ltd | CAT# G4500 | |
| Chemical compound, drug | Histidine | Merck/Sigma | CAT# H6034-100G | |
| Chemical compound, drug | *n*-Dodecyl β-maltoside (DDM-C) | Glycon Biochemicals GMBH | CAT# D97002-C-50g | |
| Chemical compound, drug | cOmplete, EDTA-free Protease Inhibitor Cocktail | Merck/Roche | CAT# 4693132001 | |
| Chemical compound, drug | AEBSF hydrochloride | Fisher Scientific/Acros | CAT# 10742885 | |
| Chemical compound, drug | Dithiothreitol | Fisher Scientific | CAT# 10592945 | |
| Chemical compound, drug | DNAse I | Merck/Roche | CAT# 10104159001 | |
| Chemical compound, drug | HisPur Ni-NTA Resin | Thermo Scientific/Pierce | CAT# 88221 | |
| Chemical compound, drug | 5/150 Superose 6 column | GE Healthcare Lifescience | CAT# 15383224 | |
| Other | Quantifoil 0.6/1 300 mesh Au holey carbon | Quantifoil Micro Tools GMBH | CAT# N1-C11nAu30-01 | Grid onto which sample is vitrified: See Methods |

*Continued on next page*

*Continued*

| Reagent type (species) or resource | Designation | Source or reference | Identifiers | Additional information |
|---|---|---|---|---|
| Other | UltrAuFoil 1.2/1.3 300 mesh Holey gold | Quantifoil Micro Tools GMBH | CAT# N1-A14nAu30-50 | Grid onto which sample is vitrified: See Methods |
| Other | UltrAuFoil 0.6/1 300 mesh Holey gold | Quantifoil Micro Tools GMBH | CAT# N1-A11nAu30-01 | Grid onto which sample is vitrified: See Methods |
| Commercial assay or kit | QuikChange II mutagenesis kit | Agilent | CAT# 200523 | |
| Sequence-based reagent | K125R-forward | This paper | PCR primers | 5'-tcgaggagatcaaaacccagagggtccgcatcg-3' |
| Sequence-based reagent | K125R-reverse | This paper | PCR primers | 5'-cgatgcggaccctctgggttttgatctcctcga-3' |
| Sequence-based reagent | K125E forward | This paper | PCR primers | 5'-gagatcaaaacccaggaggtccgcatcgaa-3' |
| Sequence-based reagent | K125E reverse | This paper | PCR primers | 5'-ttcgatgcggacctcctgggttttgatctc-3' |
| Software, algorithm | Relion | *Scheres, 2012* | https://www3.mrc-lmb.cam.ac.uk/relion | |
| Software, algorithm | UCSF Chimera | *Goddard et al., 2007* | https://www.cgl.ucsf.edu/chimera/ | |
| Software, algorithm | ModelAngelo | *Jamali et al., 2024*; *3dem, 2023* | https://github.com/3dem/model-angelo | |
| Software, algorithm | Coot | *Emsley and Cowtan, 2004* | https://www2.mrc-lmb.cam.ac.uk/personal/pemsley/coot/ | |
| Software, algorithm | PyMol | *Delano, 2002* | https://pymol.org/ | |
| Software, algorithm | Colabfold v 1.5. | *Mirdita et al., 2022* | https://colab.research.google.com/github/sokrypton/ColabFold/blob/main/AlphaFold2.ipynb | |
| Software, algorithm | CTFFIND4 | *Rohou and Grigorieff, 2015* | https://grigoriefflab.umassmed.edu/ctffind4 | |

## Mutant preparation

K125R and K125E mutations of human connexin 26 were prepared using the QuikChange II mutagenesis kit (Agilent) and the following primers: K125R forward: 5'-tcgaggagatcaaaacccagagggtccgcatcg-3', K125R reverse: 5'-cgatgcggaccctctgggttttgatctcctcga-3', K125E forward: 5'-gagatcaaaacccaggaggtccgcatcgaa-3', K125E reverse: 5'-ttcgatgcggacctcctgggttttgatctc-3' (Sigma) with the WT human connexin 26 pFast construct used for previous studies as the template for mutagenesis (*Brotherton et al., 2022*). Viruses harbouring the connexin constructs were prepared and protein expressed in *Sf9* cells.

## HeLa cell culture and transfection

HeLa DH (ECACC) cells were grown in DMEM supplemented with 10% fetal bovine serum, 50 µg/ml penicillin/streptomycin, and 3 mM CaCl$_2$. For intercellular dye transfer experiments, cells were seeded onto coverslips in six-well plates at a density of $2 \times 10^4$ cells per well. After 24 hr, the cells were transiently transfected with Cx26 constructs (*Meigh et al., 2013*) tagged at the C-terminus with a fluorescent marker (mCherry) according to the GeneJuice Transfection Reagent protocol (Merck Millipore).

## Patch clamp recording and gap junction assay

2-Deoxy-2-[(7-nitro-2,1,3-benzoxadiazol-4-yl)amino]-D-glucose, NBDG, was included at 200 µM in the patch recording fluid, which contained: K-gluconate 130 mM; KCl 10 mM; EGTA 5 mM; CaCl$_2$ 2 mM,

HEPES 10 mM, pH was adjusted to 7.3 with KOH and a resulting final osmolarity of 295 mOsm. A coverslip of cells was placed in the recording chamber and superfused with a control saline (124 mM NaCl, 3 mM KCl, 2 mM CaCl$_2$, 26 mM NaHCO$_3$, 1.25 mM NaH$_2$PO$_4$, 1 mM MgSO$_4$, and 10 mM D-glucose saturated with 95% O$_2$/5% CO$_2$, pH 7.4, PCO$_2$ 35 mmHg). Cells were imaged on a Cleverscope (MCI Neuroscience) with a Photometrics Prime camera under the control of Micromanager 1.4 software. LED illumination (Cairn Research) and an image splitter (Optosplit, Cairn Research) allowed simultaneous imaging of the mCherry-tagged Cx26 subunits and the diffusion of the NBDG into and between cells. Coupled cells for intercellular dye transfer experiments were selected based on tagged Cx26 protein expression and the presence of a gap junctional plaque, easily visible as a band of mCherry fluorescence. After establishing a gigaseal and the whole-cell mode of recording, images were collected every 10 s.

## Protein production, purification, and grid preparation

Purification of all proteins were performed as previously described (*Brotherton et al., 2022*), and briefly described here for each protein sample.

### K125E in HEPES buffer
#### Protein production and purification

*Sf9* cells harbouring the Cx26 virus were harvested at 72 hr post infection at 2500 × *g* in a Beckmann JLA 8.1000 rotor, cell pellets were snap-frozen in liquid nitrogen, and stored at –80°C until purification. All purification steps were performed on ice, or at 4°C. Cells were thawed in hypotonic lysis buffer (10 mM sodium phosphate, 10 mM NaCl, 5 mM MgCl$_2$, 1 mM DTT, pH 8.0- DNAse I, cOmplete EDTA-free Protease Inhibitor Cocktail [Roche] and AEBSF) for 30 min before breakage using a dounce homogeniser. Membranes were separated by ultracentrifugation for 1 hr at 4°C, 158,000 × *g*. After resuspending the membranes in membrane resuspension buffer (25 mM sodium phosphate, 150 mM NaCl, 5% glycerol, 1 mM DTT, pH 8.0- DNAse I, cOmplete EDTA-free Protease Inhibitor Cocktail and AEBSF) solubilisation was carried out in membrane solubilisation buffer (10 mM sodium phosphate, 300 mM NaCl, 5% glycerol, 1 mM DTT, 1% DDM [Glycon Biochemicals GMBH], pH 8.0) for 3–4 hr, and insoluble material removed by a further 1 hr ultracentrifugation at 4°C, 158,000 × *g*. Soluble protein was batch-bound to pre-equilibrated HisPur Ni-NTA resin (Thermo Scientific) overnight and then poured into an Econo-Column for subsequent manual washing and elution steps. Resin was washed with 5× CV wash buffer (10 mM sodium phosphate, 500 mM NaCl, 10 mM histidine, 5% glycerol, 1 mM DTT, 0.1% DDM, pH 8.0) before eluting hCx26 with elution buffer (10 mM sodium phosphate, 500 mM NaCl, 200 mM histidine, 5% glycerol, 1 mM DTT, 0.1% DDM, pH 8.0). hCx26-containing fractions were dialysed (20 mM HEPES, 500 mM NaCl, 5% glycerol, 1 mM DTT, 0.03% DDM, pH 8.0) overnight with thrombin at (a 1:1 wt/wt ratio). The hCx26 was then passed through a 0.2 µm filter, concentrated using a Vivaspin concentration with 100,000 MWCO and loaded onto a Superose 6 Increase 10/300 size exclusion chromatography column (GE Healthcare Lifescience) equilibrated with the same HEPES-dialysis buffer to remove thrombin. The protein was subsequently concentrated to ~3 mg/ml. The concentrated protein was then dialysed for a minimum of 3 hr prior to grid preparation against 20 mM HEPES, 250 mM NaCl, 2.5% glycerol, 5 mM DTT, 0.03% DDM, 1 mM CaCl$_2$, pH 8.0.

#### Grid preparation

Protein (3.5 mg/ml) was centrifuged at 17,200 × *g* for 5 min at 4°C. Grids (0.6/1 quantifoil AU 300) were glow discharged for 30 s at 30 mA. Vitrification of the protein in liquid ethane at –180°C was carried out with a Vitrobot MKIV with 3 µl protein per grid at 4°C, 100% humidity, blot force 10, 3 s blotting.

#### Data collection and processing

Data were collected using a Titan Krios G3 on a Falcon 3 detector. Data processing was performed in Relion 3 (*Zivanov et al., 2018*). Movies were motion corrected with MotionCor2 (*Zheng et al., 2017*) and the CTF parameters estimated with CTFfind-4.1 (*Rohou and Grigorieff, 2015*), both implemented in Relion 3. Particles were picked from selected images using the Laplacian-of-Gaussian picker, and serial rounds of 2D classifications on binned particles were used to filter out junk and poor

particles. An initial model was generated using stochastic gradient descent, and this was used for further cleaning of particles via 3D classifications. Exhaustive rounds of 3D refinement, CTF refinement, and polishing were performed on unbinned particles until no further improvement of the resolution for the Coulomb shell was gained. The resolution was estimated based on the gold standard FSC criterion (*Rosenthal and Henderson, 2003*; *Scheres, 2012*) with a soft solvent mask. All masks for processing were prepared in Chimera (*Goddard et al., 2007*; *Pettersen et al., 2004*). All processing was carried out without imposed symmetry until the final stage, where tests with C2, C3, C6, and D6 for refinement were carried out to look for improvements in resolution.

### WT in HEPES buffer

All methods are as above, with the following changes: the final dialysis prior to freezing was against 20 mM HEPES, 200 mM NaCl, 1% glycerol, 1 mM DTT, 1 mM $CaCl_2$, 0.03% DDM, pH 8.0. Freezing concentration was 3 mg/ml WT, and data collection was carried out using a Volta phase-plate.

### K125E in αCSF90 buffer

All methods are as for K125E in HEPES buffer, except for the following changes: Fractions eluted from the NiNTA containing hCx26 were dialysed overnight at 4°C against 10 mM sodium phosphate, 500 mM NaCl, 5% glycerol, 1 mM DTT, 0.03% DDM, pH 8.0. A Superose 6 Increase 5/150 size exclusion chromatography column (GE Healthcare Lifescience) was used to remove thrombin and exchange the buffer to αCSF90 buffer (70 mM NaCl, 5% glycerol, 1 mM DTT, 0.03% DDM, 80 mM $NaHCO_3$, 1.25 mM $NaH_2PO_4$, 3 mM KCl, 1 mM $MgSO_4$, 4 mM $MgCl_2$). K125E (3.4 mg/ml) was gassed with 15% $CO_2$ (3×12 s) followed by centrifugation at 17,200 × $g$ for 5 min at 4°C. Grids (0.6/1 quantifoil AU 300) were glow discharged for 1 min at 30 mA. Vitrification of the protein in liquid ethane/propane at –180°C was carried out with a Leica GP2 automated plunge freezer with 3 µl protein per grid at 4°C, 100% humidity, 7 s blotting without sensor-blot in a 15% $CO_2$ atmosphere. Data were collected using a GATAN K3 detector in super-resolution mode and were processed using Relion 4.

### K125R in αCSF90 buffer

All methods are as for K125E in αCSF90 buffer, except for the following changes: Grids (1.2/1.3 UltrAuFoil Au300) were glow discharged at 30 mA for 30 s. Vitrification of the protein in liquid ethane at –160°C was carried out with a Vitrobot with 3 µl protein per grid at 4°C, 100% humidity, 3 s blotting (force 10, 1 blot, skip transfer) in a 15% $CO_2$ atmosphere. Data were collected using a K3 detector in counting bin 1 mode. Data processing was carried out in Relion 4.

### LMNG$_{90}$ hCx26 WT

Preparation of LMNG$_{90}$ hCx26 WT protein was carried out as for K125E in αCSF90 buffer, with the following changes: *Sf9* cells were lysed in αCSF90 buffer (70 mM NaCl, 5% glycerol, 1 mM DTT, 80 mM $NaHCO_3$, 1.25 mM $NaH_2PO_4$, 3 mM KCl, 1 mM $MgSO_4$, 4 mM $MgCl_2$, pH corrected to 7.4 by addition of $CO_2$) and membranes were resuspended in (110 mM NaCl, 5% glycerol, 1 mM DTT, 80 mM $NaHCO_3$, 1.25 mM $NaH_2PO_4$, 3 mM KCl, 1 mM $MgSO_4$, 4 mM $MgCl_2$, pH corrected to 7.4 by addition of $CO_2$) and solubilised in (500 mM NaCl, 5% glycerol, 1 mM DTT, 80 mM $NaHCO_3$, 1.25 mM $NaH_2PO_4$, 3 mM KCl, 1 mM $MgSO_4$, 4 mM $MgCl_2$, pH corrected to 7.4 by addition of $CO_2$). Samples were taken periodically to check the pH, and re-adjusted by further addition of $CO_2$ when necessary to keep the pH constant. Wash buffer for NiNTA resin (500 mM NaCl, 10 mM histidine, 5% glycerol, 1 mM DTT, 80 mM $NaHCO_3$, 1.25 mM $NaH_2PO_4$, 3 mM KCl, 1 mM $MgSO_4$, 4 mM $MgCl_2$, pH corrected to 7.4 by addition of $CO_2$) and elution buffer (500 mM NaCl, 200 mM histidine, 5% glycerol, 1 mM DTT, 80 mM $NaHCO_3$, 1.25 mM $NaH_2PO_4$, 3 mM KCl, 1 mM $MgSO_4$, 4 mM $MgCl_2$, pH corrected to 7.4 by addition of $CO_2$) were prepared and the pH checked just prior to use, to ensure no drifting of pH before interaction with the connexin. Selected fractions eluted from NiNTA were dialysed against (500 mM NaCl, 5% glycerol, 1 mM DTT, 80 mM $NaHCO_3$, 1.25 mM $NaH_2PO_4$, 3 mM KCl, 1 mM $MgSO_4$, 4 mM $MgCl_2$, pH corrected to 7.4 by addition of $CO_2$). The final size exclusion step was performed in αCSF90 buffer (70 mM NaCl, 5% glycerol, 1 mM DTT, 0.03% DDM, 80 mM $NaHCO_3$, 1.25 mM $NaH_2PO_4$, 3 mM KCl, 1 mM $MgSO_4$, 4 mM $MgCl_2$) without additional $CO_2$. The concentrated, pooled samples were gassed to pH to 7.4 both prior to freezing as described previously (*Brotherton et al., 2022*). Vitrification was carried out at 3.7 mg/ml on 0.6/1 UltrAuFoil grids

using the method described for K125R in αCSF90 buffer. Data were collected using the K3 detector in super-resolution bin 2 mode.

### Particle subtraction and masked classification focussed on the hemichannel

Hemichannel classifications with C6 imposed symmetry was carried out as described previously (*Brotherton et al., 2022*). The particles from the top class were unsubtracted, and the particles were refined with C6 symmetry, using a hemichannel mask and limited angular sampling. Local resolution estimation was carried out in Relion.

### Particle subtraction and masked classification focussed on a single subunit

Following particle expansion with D6 symmetry and particle subtraction with a mask encompassing a single subunit, masked, fixed angle classification was carried out in Relion 4. Following unsubtraction of particles, refinement of the particle positions was carried out as above, with a hemichannel mask.

## Model building and refinement

Model building was carried out in Coot (*Emsley and Cowtan, 2004*) with real space refinement in Phenix (*Liebschner et al., 2019*) using maps that had been sharpened using model-free local sharpening in Phenix. For the $LMNG_{90}$ hemichannel-based classification two maps were selected for refinement. The first of these ($LMNG-N_{Const}$) was chosen because the density of the cytoplasmic region was the most defined. The second ($LMNG-N_{Flex}$) was chosen as the highest resolution map with TM2 in the most diverse position. A similar selection was made for the single subunit-based classification. In building the cytoplasmic region of the protein reference was made to both ModelAngelo (*Jamali et al., 2024*) and AlphaFold2 (*Jumper and Hassabis, 2022*). AlphaFold2 structures were created with Colabfold (*Mirdita et al., 2022*) or downloaded from the EBI (*Varadi et al., 2022*).

## Structural analysis

All structural images shown in this paper were generated in PyMol (*Delano, 2002*) or Chimera (*Goddard et al., 2007*; *Pettersen et al., 2004*). Superpositions were carried out in Chimera such that only matching $C_\alpha$ pairs within 2 Å after superposition were included in the matrix calculation.

## Acknowledgements

We thank the Leverhulme Trust (RPG-2015-090) and MRC (MR/P010393/1) for support. We acknowledge the Midlands Regional Cryo-EM Facility, hosted at the Warwick Advanced Bioimaging Research Technology Platform, for use of the JEOL 2100Plus, and the Midlands Regional Cryo-EM Facility, hosted at Leicester Institute of Structural and Chemical Biology for use of the FEI Titan Krios G3 and associated facilities, supported by MRC award reference MC_PC_17136. We thank Dr TJ Ragan for help with cryo-EM. We are grateful to the technical support in the School of Life Sciences, University of Warwick.

## Additional information

### Funding

| Funder | Grant reference number | Author |
| --- | --- | --- |
| Leverhulme Trust | RPG-2015-090 | Deborah H Brotherton<br>Nicholas Dale<br>Alexander David Cameron |
| Medical Research Council | MR/P010393/1 | Deborah H Brotherton<br>Sarbjit Nijjar<br>Nicholas Dale<br>Alexander David Cameron |
| Medical Research Council | MC_PC_17136 | Christos G Savva |

| Funder | Grant reference number | Author |
|---|---|---|

The funders had no role in study design, data collection and interpretation, or the decision to submit the work for publication.

## Author contributions

Deborah H Brotherton, Data curation, Formal analysis, Investigation, Writing – original draft, Protein purification, Grid preparation, Data collection, Processing and structure refinement; Sarbjit Nijjar, Data curation, Investigation, Writing – review and editing, Dye transfer assays; Christos G Savva, Investigation, Writing – review and editing, Data collection; Nicholas Dale, Conceptualization, Data curation, Formal analysis, Supervision, Investigation, Writing – original draft; Alexander David Cameron, Conceptualization, Data curation, Formal analysis, Supervision, Funding acquisition, Investigation, Writing – original draft

## Author ORCIDs

Nicholas Dale  http://orcid.org/0000-0003-2196-2949
Alexander David Cameron  https://orcid.org/0000-0001-8776-3518

Reviewer #1 (Public Review): https://doi.org/10.7554/eLife.93686.3.sa1
Reviewer #2 (Public Review): https://doi.org/10.7554/eLife.93686.3.sa2
Reviewer #3 (Public Review): https://doi.org/10.7554/eLife.93686.3.sa3
Author response https://doi.org/10.7554/eLife.93686.3.sa4

# Additional files

## Supplementary files

• MDAR checklist

## Data availability

Cryo-EM density maps have been deposited in the Electron Microscopy Data Bank (EMDB) under accession numbers EMD-18290 (K125E90), EMD-18295 (K125R90), EMD-18296 (K125EHEPES), EMD-18297 (WTHEPES), EMD-18291 (LMNG-NConst), EMD-18292 (LMNG-NFlex), EMD-18293 (LMNG-NConst-mon), EMD-18294 (LMNG-NFlex-mon). Structure models have been deposited in the RCSB Protein Data Bank under accession numbers 8Q9Z, 8QA1, 8QA0, 8QA2, 8QA3. *Figure 4—source data 1* contain the numerical data used to generate the figure.

The following datasets were generated:

| Author(s) | Year | Dataset title | Dataset URL | Database and Identifier |
|---|---|---|---|---|
| Brotherton DH, Cameron AD | 2024 | Cryo-EM structure of Cx26 solubilised in LMNG - hemichannel classification - NConst conformation | https://www.rcsb.org/structure/8QA0 | RCSB Protein Data Bank, 8QA0 |
| Brotherton DH, Cameron AD | 2024 | Cryo-EM structure of Cx26 solubilised in LMNG - Hemichannel classification NFlex conformation | https://www.rcsb.org/structure/8QA1 | RCSB Protein Data Bank, 8QA1 |
| Brotherton DH, Cameron AD | 2024 | Cryo-EM structure of Cx26 solubilised in LMNG: classification on subunit A; Nconst-mon conformation | https://www.rcsb.org/structure/8QA2 | RCSB Protein Data Bank, 8QA2 |
| Brotherton DH, Cameron AD | 2024 | Cryo-EM structure of Cx26 solubilised in LMNG: classification on subunit A; NFlex conformation | https://www.rcsb.org/structure/8QA3 | RCSB Protein Data Bank, 8QA3 |

*Continued*

| Author(s) | Year | Dataset title | Dataset URL | Database and Identifier |
|---|---|---|---|---|
| Brotherton DH, Cameron AD | 2024 | Cryo-EM structure of Cx26 gap junction K125E mutant in bicarbonate buffer (classification on hemichannel) | https://www.rcsb.org/structure/8Q9Z | RCSB Protein Data Bank, 8Q9Z |
| Brotherton DH, Savva CG, Cameron AD | 2024 | Cryo-EM structure of Cx26 gap junction K125E mutant in bicarbonate buffer (classification on hemichannel) | https://www.ebi.ac.uk/emdb/EMD-18290 | Electron Microscopy Data Bank, EMD-18290 |
| Brotherton DH, Savva CG, Cameron AD | 2024 | Cryo-EM structure of Cx26 solubilised in LMNG - hemichannel classification - NConst conformation | https://www.ebi.ac.uk/emdb/EMD-18291 | Electron Microscopy Data Bank, EMD-18291 |
| Brotherton DH, Savva CG, Cameron AD | 2024 | Cryo-EM structure of Cx26 solubilised in LMNG - Hemichannel classification NFlex conformation | https://www.ebi.ac.uk/emdb/EMD-18292 | Electron Microscopy Data Bank, EMD-18292 |
| Brotherton DH, Savva CG, Cameron AD | 2024 | Cryo-EM structure of Cx26 solubilised in LMNG: classification on subunit A; Nconst-mon conformation | https://www.ebi.ac.uk/emdb/EMD-18293 | Electron Microscopy Data Bank, EMD-18293 |
| Brotherton DH, Savva CG, Cameron AD | 2024 | Cryo-EM structure of Cx26 solubilised in LMNG: classification on subunit A; NFlex conformation | https://www.ebi.ac.uk/emdb/EMD-18294 | Electron Microscopy Data Bank, EMD-18294 |
| Brotherton DH, Savva CG, Cameron AD | 2024 | Cryo-EM reconstruction of Cx26 gap junction K125R mutant (D6 symmetry) | https://www.ebi.ac.uk/emdb/EMD-18295 | Electron Microscopy Data Bank, EMD-18295 |
| Brotherton DH, Savva CG, Cameron AD | 2024 | Cryo-EM reconstruction of Cx26 gap junction K125E mutant in HEPES buffer | https://www.ebi.ac.uk/emdb/EMD-18296 | Electron Microscopy Data Bank, EMD-18296 |
| Brotherton DH, Savva CG, Cameron AD | 2024 | Cryo-EM reconstruction of Cx26 gap junction WT in HEPES buffer | https://www.ebi.ac.uk/emdb/EMD-18297 | Electron Microscopy Data Bank, EMD-18297 |

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
