## [Editor Report · eLife assessment]

This study presents **valuable** new structures of a carbamylation-mimetic K125E mutant of the Cx26 gap junction channel uncovering the cytoplasmic loop structure and information about the closed state of the channel. The cryo-EM maps are in high quality and serve as strong foundations for dissecting the gating mechanism by CO2, providing **convincing** evidence in support of a mechanism where CO2-mediated carbamylation of Lys125 shifts the conformational equilibrium towards a state where the N-terminus occludes the pore of the channel. This information will be of interest to biochemists, cell biologists and biophysicists interested in the function of gap-junction channels in health and disease.

---

## [Referee Report · Reviewer #1 (Public Review)]

Gap junction channels establish gated intercellular conduits that allow the diffusion of solutes between two cells. Hexameric connexin26 (Cx26) hemichannels are closed under basal conditions and open in response to CO2. In contrast, when forming a dodecameric gap-junction, channels are open under basal conditions and close with increased CO2 levels. Previous experiments have implicated Cx26 residue K125 in the gating mechanism by CO2, which is thought to become carbamylated by CO2. Carbamylation is a labile post-translational modification that confers negative charge to the K125 side chain. How the introduction of a negative charge at K125 causes a change in gating is unclear, but it has been proposed that carbamylated K125 forms a salt bridge with the side chain at R104, causing a conformational change in the channel. It is also unclear how overall gating is controlled by changes in CO2, since there is significant variability between structures of gap-junction channels and the cytoplasmic domain is generally poorly resolved. Structures of WT Cx26 gap-junction channels determined in the presence of various concentrations of CO2 have suggested that the cytoplasmatic N-terminus changes conformation depending on the concentration of the gas, occluding the pore when CO2 levels are high.

In the present manuscript, Deborah H. Brotherton and collaborators use an intercellular dye-transfer assay to show that Cx26 gap-junction channels containing the K125E mutation, which mimics carbamylation caused by CO2, is constitutively closed even at CO2 concentrations where WT channels are open. Several cryo-EM structures of WT and mutant Cx26 gap junction channels were determined at various conditions and using classification procedures that extracted more than one structural class from some of the datasets. Together, the features on each of the different structures are generally consistent with previously obtained structures at different CO2 concentrations and support the mechanism that is proposed in the manuscript. The most populated class for K125E channels determined at high CO2 shows a pore that is constricted by the N-terminus, and a cytoplasmic region that was better resolved than in WT channels, suggesting increased stability. The K125E structure closely resembles one of the two major classes obtained for WT channels at high CO2. These findings support the hypothesis that the K125E mutation biases channels towards the closed state, while WT channels are in an equilibrium between open and closed states even in the presence of high CO2. Consistently, a structure of K125E obtained in the absence of CO2 appeared to also represent a closed state but at a lower resolution, suggesting that CO2 has other effects on the channel beyond carbamylation of K125 that also contribute to stabilizing the closed state. Structures determined for K125R channels, which are constitutively open because arginine cannot be carbamylated, and would be predicted to represent open states, yielded apparently inconclusive results.

A non-protein density was found to be trapped inside the pore in all structures obtained using both DDM and LMNG detergents, suggesting that the density represents a lipid rather than a detergent molecule. It is thought that the lipid could contribute to the process of gating, but this remains speculative. The cytoplasmic region in the tentatively closed structural class of the WT channel obtained using LMNG was better resolved. An additional portion of the cytoplasmic face could be resolved by focusing classification on a single subunit, which had a conformation that resembled the AlphaFold prediction. However, this single-subunit conformation was incompatible with a C6-symmetric arrangement. Together, the results suggest that the identified states of the channel represent open states and closed states resulting from interaction with CO2. Therefore, the observed conformational changes illuminate a possible structural mechanism for channel gating in response to CO2.

---

## [Referee Report · Reviewer #2 (Public Review)]

Summary:

The manuscript by Brotherton et al. describes a structural study of connexin-26 (Cx26) gap junction channel mutant K125E, which is designed to mimic the CO2-inhibited form of the channel. In the wild-type Cx26, exposure to CO2 is presumed to close the channel through carbamylation of the redeye K125. The authors mutated K125 to a negatively charged residue to mimic this effect and observed by cryo-EM analysis of the mutated channel that the pore of the channel is constricted. The authors were able to observe conformations of the channel with resolved density for the cytoplasmic loop (in which K125 is located). Based on the observed conformations and on the position of the N-terminal helix, which is involved in channel gating and in controlling the size of the pore, the authors propose the mechanisms of Cx26 regulation.

Strengths:

This is a very interesting and timely study, and the observations provide a lot of new information on connexin channel regulation. The authors use the state of the art cryo-EM analysis and 3D classification approaches to tease out the conformations of the channel that can be interpreted as "inhibited", with important implications for our understanding of how the conformations of the connexin channels controlled.

Weaknesses:

The revised version of the manuscript is improved, and the authors have addressed the review comments/criticisms in a satisfactory manner.

---

## [Referee Report · Reviewer #3 (Public Review)]

Summary:

The mechanism underlying the well-documented CO2-regulated activity of connexin 26 (Cx26) remains poorly understood. This is largely due to the labile nature of CO2-mediated carbamylation, making it challenging to visualize the effects of this reversible posttranslational modification. This paper by Brotherton et al. aims to address this gap by providing structural insights through cryo-EM structures of a carbamylation-mimetic mutant of the gap junction protein.

Strength:

The combination of the mutation, elevated PCO2, and the use of LMNG detergent resulted in high-resolution maps that revealed, for the first time, the structure of the cytoplasmic loop between transmembrane helix (TM) 2 and 3.

Weaknesses:

While the structure of the TM2-TM3 loop may suggest a mechanism for stabilizing the closed conformation, the EM density is not strong enough to support direct interaction with carbamylated or mutated K125.

Overall, the cryo-EM structures presented in this study support their proposing mechanism in which carbamylation at K125 promotes Cx26 gap junction closure. Through careful control of the pH and PCO2 for each cryo-EM sample, the current study substantiated that the more closed conformation observed in high PCO2 is independent of pH but likely triggered by carbamylation. This was unclear from their prior cryo-EM map of wildtype Cx26 at high PCO2.

While the new structures successfully visualize the TM2-TM3 loop, which likely plays significant roles in CO2-regulated Cx26 activity, further studies are necessary to understand the underlying mechanism. For instance, the current study lacks explanation regarding what propels the movement of the N-terminal helix, how carbamylated K125 interacts with the TM2-TM3 loop, the importance of the lipids visualized in the map, or the reason why gap junctions are constitutively open while hemichannels are closed under normal PCO2 levels

---

## [Author Response]

The following is the authors’ response to the previous reviews.

**Public Reviews:**

**Reviewer #1 (Public Review):**
Gap junction channels establish gated intercellular conduits that allow the diffusion of solutes between two cells. Hexameric connexin26 (Cx26) hemichannels are closed under basal conditions and open in response to CO2. In contrast, when forming a dodecameric gapjunction, channels are open under basal conditions and close with increased CO2 levels. Previous experiments have implicated Cx26 residue K125 in the gating mechanism by CO2, which is thought to become carbamylated by CO2. Carbamylation is a labile post-translational modification that confers negative charge to the K125 side chain. How the introduction of a negative charge at K125 causes a change in gating is unclear, but it has been proposed that carbamylated K125 forms a salt bridge with the side chain at R104, causing a conformational change in the channel. It is also unclear how overall gating is controlled by changes in CO2, since there is significant variability between structures of gap-junction channels and the cytoplasmic domain is generally poorly resolved. Structures of WT Cx26 gap-junction channels determined in the presence of various concentrations of CO2 have suggested that the cytoplasmatic N-terminus changes conformation depending on the concentration of the gas, occluding the pore when CO2 levels are high.In the present manuscript, Deborah H. Brotherton and collaborators use an intercellular dyetransfer assay to show that Cx26 gap-junction channels containing the K125E mutation, which mimics carbamylation caused by CO2, is constitutively closed even at CO2 concentrations where WT channels are open. Several cryo-EM structures of WT and mutant Cx26 gap junction channels were determined at various conditions and using classification procedures that extracted more than one structural class from some of the datasets. Together, the features on each of the different structures are generally consistent with previously obtained structures at different CO2 concentrations and support the mechanism that is proposed in the manuscript. The most populated class for K125E channels determined at high CO2 shows a pore that is constricted by the N-terminus, and a cytoplasmic region that was better resolved than in WT channels, suggesting increased stability. The K125E structure closely resembles one of the two major classes obtained for WT channels at high CO2. These findings support the hypothesis that the K125E mutation biases channels towards the closed state, while WT channels are in an equilibrium between open and closed states even in the presence of high CO2. Consistently, a structure of K125E obtained in the absence of CO2 appeared to also represent a closed state but at lower resolution, suggesting that CO2 has other effects on the channel beyond carbamylation of K125 that also contribute to stabilizing the closed state. Structures determined for K125R channels, which are constitutively open because arginine cannot be carbamylated, and would be predicted to represent open states, yielded apparently inconclusive results.A non-protein density was found to be trapped inside the pore in all structures obtained using both DDM and LMNG detergents, suggesting that the density represents a lipid rather than a detergent molecule. It is thought that the lipid could contribute to the process of gating, but this remains speculative. The cytoplasmic region in the tentatively closed structural class of the WT channel obtained using LMNG was better resolved. An additional portion of the cytoplasmic face could be resolved by focusing classification on a single subunit, which had a conformation that resembled the AlphaFold prediction. However, this single-subunit conformation was incompatible with a C6-symmetric arrangement. Together, the results suggest that the identified states of the channel represent open states and closed states resulting from interaction with CO2. Therefore, the observed conformational changes illuminate a possible structural mechanism for channel gating in response to CO2.Some of the discussion involving comparisons with structures of other gap junction channels are relatively hard to follow as currently written, especially for a general readership. Also, no additional functional experiments are carried out to test any of the hypotheses arising from the data. However, structures were determined in multiple conditions, with results that were consistent with the main hypothesis of the manuscript. No discussion is provided, even if speculative, to explain the difference in behavior between hemichannels and gap junction channels. Also, no attempt was made to measure the dimensions of the pore, which is relevant because of the importance of identifying if the structures indeed represent open or closed states of the channel.

We have considerably revised the manuscript in an attempt to make it more tractable. We respond to the individual comments below.

**Reviewer #2 (Public Review):**
Summary:The manuscript by Brotherton et al. describes a structural study of connexin-26 (Cx26) gap junction channel mutant K125E, which is designed to mimic the CO2-inhibited form of the channel. In the wild-type Cx26, exposure to CO2 is presumed to close the channel through carbamylation of the residue K125. The authors mutated K125 to a negatively charged residue to mimic this effect, and they observed by cryo-EM analysis of the mutated channel that the pore of the channel is constricted. The authors were able to observe conformations of the channel with resolved density for the cytoplasmic loop (in which K125 is located). Based on the observed conformations and on the position of the N-terminal helix, which is involved in channel gating and in controlling the size of the pore, the authors propose the mechanisms of Cx26 regulation.Strengths:This is a very interesting and timely study, and the observations provide a lot of new information on connexin channel regulation. The authors use the state of the art cryo-EM analysis and 3D classification approaches to tease out the conformations of the channel that can be interpreted as "inhibited", with important implications for our understanding of how the conformations of the connexin channels controlled.Weaknesses:My fundamental question to the premise of this study is: to what extent can K125 carbamylation by recapitulated by a simple K125E mutation? Lysine has a large side chain, and its carbamylation would make it even slightly larger. While the authors make a compelling case for E125-induced conformational changes focusing primarily on the negative charge, I wonder whether they considered the extent to which their observation with this mutant may translate to the carbamoylated lysine in the wild-type Cx26, considering not only the charge but also the size of the modified side-chain.

This is an important point. We agree that the difference in size will have a different effect on the structure. For kinases, aspartate or glutamate are often used as mimics of phosphorylated serine or threonine and these will have the same issues. The fact that we cannot resolve the relevant side-chains in the density may be indicative that the mutation doesn’t give the whole story. It may be able to shift the equilibrium towards the closed conformation, but not stably trap the molecule in that conformation. We include a comment to this effect in the revised manuscript.

**Reviewer #3 (Public Review):**
Summary:The mechanism underlying the well-documented CO2-regulated activity of connexin 26 (Cx26) remains poorly understood. This is largely due to the labile nature of CO2-mediated carbamylation, making it challenging to visualize the effects of this reversible posttranslational modification. This paper by Brotherton et al. aims to address this gap by providing structural insights through cryo-EM structures of a carbamylation-mimetic mutant of the gap junction protein.Strengths:The combination of the mutation, elevated PCO2, and the use of LMNG detergent resulted in high-resolution maps that revealed, for the first time, the structure of the cytoplasmic loop between transmembrane helix (TM) 2 and 3.Weaknesses:The presented maps merely reinforce their previous findings, wherein wildtype Cx26 favored a closed conformation in the presence of high PCO2. While the structure of the TM2-TM3 loop may suggest a mechanism for stabilizing the closed conformation, no experimental data was provided to support this mechanism. Additionally, the cryo-EM maps were not effectively presented, making it difficult for readers to grasp the message.

We have extensively revised the manuscript so that the novelty of this study is more apparent.There are three major points

(1) The carbamylation mimetic pushes the conformation towards the closed conformation. Previously we just showed that CO2 pushes the conformation towards this conformation. Though we could show this was not due to pH, and could speculate this was due to carbamylation as suggested by previous mutagenesis studies, our data did not provide any mechanism whereby Lys125 was involved.

(2) In going from the open to closed conformations, not only is a conformational change in TM2 involved, as we saw previously, but also a conformational change in TM1, the linker to the N-terminus and the cytoplasmic loop. Thus there is a clear connection between Lys125 and the conformation of the pore-closing N-terminus.

(3) We observe for the first time in any connexin structure, density for the cytoplasmic loop. Since this loop is important in regulation, knowing how it might influence the positions of the transmembrane helices is important information if we are to understand how connexins can be regulated.

**Reviewing Editor:**
The reviewers have agreed on a list of suggested revisions that would improve the eLife assessment if implemented, which are as follows:(1) For completeness, Figure 1 could be supplied with an example of how the experiment would look like in the presence of CO2 - for the wild-type and for the K125E mutant. presumably for the wild-type this has been done previously in exactly this assay format, but this control would be an important part of characterization for the mutant. Page 4, lines 105106; "unsurprisingly, Cx26K125E gap junctions remain closed at a PCO2 of 55 mmHg." The data should be presented in the manuscript.

We have now included the data with a PCO2 of 55mmH. This is now Figure 4 in our revised manuscript.

(2) Would AlphaFold predictions show any interpretable differences in the E125 mutant, compared to the K125 (the wild-type)?

We tried this in response to the reviewer’s suggestion. We did not see any interpretable differences. In general AlphaFold is not recognised as giving meaningful information around point mutations.

(3) The K125R mutant appears to be a more effective control for extracting significant features from the K125E maps. Given that the use of a buffer containing high PCO2 is essential for obtaining high-resolution maps, wildtype Cx26 is unsuitable as an appropriate control. The K125R map, obtained at a high resolution (2.1Å), supports its suitability as a robust control.

Though we are unsure what the referee is referring to here, we have rewritten this section and compare against the K125R map (figure 5a) as well as that derived from the wild-type protein. The important point is that the K125E mutant, causes a structural change that is consistent with the closure of the gap junctions that we observe in the dye-transfer assays.

(4) Likewise, the rationale for using wildtype Cx26 maps obtained in DDM is unclear. Wildtype Cx26 seems to yield much better cryo-EM maps in LMNG. We suggest focusing the manuscript on the higher-quality maps, and providing supporting information from the DDM maps to discuss consistency between observations and the likely possibility that the nonprotein density in the pore is lipid and not detergent.

The rationale for comparing the mutants against the wt Cx26 maps obtained in DDM was because the mutants were also solubilised in DDM. However, taking the lead from the referees’ comments, we have now rewritten the manuscript so that we first focus on the data we obtain from protein solubilised in LMNG. We feel this makes our message much clearer.

(5) In general, the rationale for utilizing cryo-EM maps with the entire selected particles is unclear. Although the overall resolutions may slightly improve in this approach, the regions of interest, such as the N-terminus and the cytoplasmic loop, appear to be better ordered afer further classifications. The paper would be more comprehensible if it focuses solely on the classes representing the pore-constricting N-terminus (PCN) and the pore-open flexible Nterminus (POFN) conformations. Also, the nomenclatures used in the manuscript, such as "WT90-Class1", "K125E90-1", "LMNG90-class1", "LMNG90-mon-pcn" are confusing.LMNG90s are also wildtype; K125E-90-1 is in Class1 for this mutant and is similar to WT90Class2, which represents the PCN conformation. More consistent and intuitive nomenclatures would be helpful.

We agree with the referees’ comments. This should now be clearer with our rewritten manuscript where we have simplified this considerably. We now call the conformations NConst (N-terminus defined and constricting the pore) and NFlex (N-terminus not visible) and keep this consistent throughout.

(6) A potential salt bridge between the carbamylated K125 and R104 is proposed to account for the prevalence of Class-1 (i.e., PCN) in the majority of cryo-EM particles. However, the side chain densities are not well-defined, suggesting that such an interaction may not be strong enough to trap Cx26 in a closed conformation. Furthermore, the absence of experimental data to support this mechanism makes it unclear how likely this mechanism may be. Combining simple mutagenesis, such as R104E, with a dye transfer assay could offer support for this mechanism. Are there any published experimental results that could help address this question without the need for additional experimental work? Alternatively, as acknowledged in the discussion, this mechanism may be deemed as an "over-simplification." What is an alternative mechanism?

R104 has been mutated to alanine in gap junctions and tested in a dye transfer assay as now mentioned in the text (Nijar et al, J Physiol 2021) supporting this role. In hemichannels R104 has been mutated to both alanine and glutamate and tested through dye loading assays Meigh et al, eLife 2013. Also in hemichannels R104 and K125 have been mutated to cysteines allowing them to be cross-linked through a disulphide bond. This mutant responds to a change in redox potential in a similar way to which the wild type protein responds to CO2 (Meigh et al, Open Biol 2015). Therefore, there is no doubt that the residues are important for the mechanism and the salt-bridge interaction seems a plausible mechanism to reconcile the mutagenesis data, however we cannot be sure that there are not other interactions involved that are necessary for closure. This information has now been included in the text.

(7) The cryo-EM maps presented in the manuscript propose that gap junctions are constitutively open under normal PCO2 as the flexible N-terminus clears the solute permeation pathway in the middle of the channel. However, hemichannels appear to be closed under normal PCO2. It is puzzling how gap junctions can open when hemichannels are closed under normal PCO2 conditions. If this question has been addressed in previous studies, the underlying mechanism should be explicitly described in the introduction. If it remains an open question, differences in the opening mechanisms between hemichannels and gap junctions should be investigated.

We suspect this is due to the difference in flexibility of gap junctions relative to hemichannels. However, a discussion of this is beyond this paper and would be complete speculation based on hemichannel structures of other connexins, performed in different buffering systems. There are no high resolution structures of Cx26 hemichannels.

(8) A mystery density likely representing a lipid is abruptly introduced, but the significance of this discovery is unclear. It is hard to place the lipid on Figure S6 in the wider context of everything else that is discussed in the text. It would be helpful for readers if a figure were provided to show where the density is located in relation to all the other regions that are extensively discussed in the text.

In the revised text this section has been completely rewritten. We have now include a more informative view in a new figure (Figure 1 – figure supplement 3).

(9) Including and displaying even tentative pore-diameter measurements for the different states - this would be helpful for readers and provide a more direct visual cue as to the difference between open and closed states.

We have purposely avoided giving precise measurements to the pore-diameter, since this depends on how we model the N-terminus. The first three residues are difficult to model into the density without causing stearic clashes with the neighbouring subunits.

(10) Given that no additional experiments for channel function were carried out, it would be useful if to provide a more detailed discussion of additional mutagenesis results from the literature that are related to the experimental results presented.

We have amplified this in the discussion (see answer to point 6).

The reviewers also agreed that improvements in the presentation of the data would strengthen the manuscript. Here is a summary list of suggestions by reviewers aimed at helping improve how the data is presented:

(1) Why is the pipette bright green in the top image, but rather weakly green in the bottom image in Figure 1 - is this the case for all images?

(Now figure 4) This depends on whether the pipette was in the focal plane of view or not. The important point of these images is the difference in intensity of the donor vs the recipient cell. The graphs in figure 4c illustrate clearly the difference between the wild-type and the mutant gap junctions.

(2) In figures 2-5, labels would help a lot in understanding what is shown - while the legends do provide the information on what is presented, it would help the reader to see the models/maps with labels directly in the panel. For example, Figure 2a/b - just indicating "WT90 Cx26" in pink and "K125E90" in blue directly in the panel would reduce the work for the reader.

We have extensively modified the labels in the figures to address this issue.

(3) Figure 4 - magenta and pink are fairly close, and to avoid confusion it might be useful to use a different color selection. This is especially true when structures are overlayed, as in this figure - the presentation becomes rather complicated, so the less confusion the color code can introduce, the better.

(Now Figure 2) We have now changed pink to blue.

(4) Figure 5 - a remarkably under-labelled figure.

Now added labels.

(5) Figure 6 - it would be interesting to add a comparison to Cx32 here as well for completeness, since the structure has been published in the meantime.

Cx32 has now been included.

(6) Figure 7 - please add equivalent labels on both sides of the model, left and right. Add the connecting lines for all of the tubes TM helices - this will help trace the structural elements shown. The legend does not quite explain the colors.

We have modified the figure as suggested and explained the colours in the legend.

(8) Fig.1 legend; Unclear what mCherry fluorescence represents. State that Cx26 was expressed as a translational fusion with mCherry.

Now figure 4. We have now written “Montages each showing bright field DIC image of HeLa cells with mCherry fluorescence corresponding to the Cx26K125E-mCherry fusion superimposed(leftmost image) and the permeation of NBDG from the recorded cell to coupled cells.”

(9) Fig. 3 b; Show R104 in the figure. Also E129-R98/R99 interaction is hard to acknowledge from the figure. It seems that the side chain density of E129 is not strong enough to support the modeled orientation.

This is now Figure 1c. While the density in this region is sufficient to be confident of the main chain, we agree that the side chain density for the E129-R98/R99 interaction is not sufficiently clear to draw attention to and have removed the associated comment from the figure legend. The density is focussed on the linker between TM1 and the N-terminus and the KVRIEG motif. We prefer to omit R104, in order to keep the focus on this region. As described in the manuscript, the density for the R104 side chain is poor.

(10) Fig. 3 c; Label the N-terminus and KVRIEG motif in the figure.

Now Figure 1b. We have labelled the N-terminus. The KVRIEG motif is not visible in this map.

(11) Page 9, lines 246-248; Restate, "We note, however, density near to Lys125, between Ser19 in the TM1-N-term linker, Tyr212 of TM4 and Tyr97 on TM3 of the neighbouring subunit, which we have been unable to explain with our modelling."

We have reworded this.

(12) Page 14, line 399; Patch clamp recording is not included in the manuscript.

Patch clamp recordings were used to introduce dye into the donor cell.

(13) On the same Figure 2, clashes are mentioned but these are hard to appreciate in any of the figures shown. Perhaps would be useful to include an inset showing this.

We have modified Figure 2b slightly and added an explanation to highlight the clash. It is slightly confusing because the residues involved belong to neighbouring subunits.

(14) The discussion related to Figure 6 is very hard to follow for readers who are not familiar with the context of abbreviations included on the figure labels. This figure could be improved to allow a general readership to identify more clearly each of the features and structural differences that are discussed in the text.

We have extensively changed the text and updated the labels on the figure to make it much easier for the reader to follow.

Below, you can find the individual reviews by each of the three reviewers.
**Reviewer #1 (Recommendations For The Authors):**
(1) In Figure 2d-e, the text discusses differences between K125E 90-1 and WT 90-class2 (7QEW), yet the figure compares K125E with 7QEQ. I suggest including a figure panel with a comparison between the two structures discussed in the manuscript text.

This has been changed in the revised manuscript.

Other comments have been addressed above.